# GAK and PRKCD are positive regulators of PRKN-independent mitophagy

Michael J. Munson [1,2,3 ✉], Benan J. Mathai[1,2,7], Matthew Yoke Wui Ng[1,2,7], Laura Trachsel-Moncho [1,2,8], Laura R. de la Ballina [1,2,8], Sebastian W. Schultz[2,4], Yahyah Aman[5], Alf H. Lystad [1,2], Sakshi Singh[1,2], Sachin Singh[2,6], Jørgen Wesche [2,6], Evandro F. Fang [5] & Anne Simonsen [1,2,4 ✉]

The mechanisms involved in programmed or damage-induced removal of mitochondria by mitophagy remains elusive. Here, we have screened for regulators of PRKN-independent mitophagy using an siRNA library targeting 197 proteins containing lipid interacting domains. We identify Cyclin G-associated kinase (GAK) and Protein Kinase C Delta (PRKCD) as regulators of PRKN-independent mitophagy, with both being dispensable for PRKN-dependent mitophagy and starvation-induced autophagy. We demonstrate that the kinase activity of both GAK and PRKCD are required for efficient mitophagy in vitro, that PRKCD is present on mitochondria, and that PRKCD facilitates recruitment of ULK1/ATG13 to early autophagic structures. Importantly, we demonstrate in vivo relevance for both kinases in the regulation of basal mitophagy. Knockdown of GAK homologue (*gakh-1*) in *C. elegans* or knockout of PRKCD homologues in zebrafish led to significant inhibition of basal mitophagy, highlighting the evolutionary relevance of these kinases in mitophagy regulation.

[1] Division of Biochemistry, Department of Molecular Medicine, Institute of Basic Medical Sciences, University of Oslo, N-0372 Oslo, Norway. [2] Centre for Cancer Cell Reprogramming, Institute of Clinical Medicine, Faculty of Medicine, University of Oslo, N-0316 Oslo, Norway. [3] Advanced Drug Delivery, Pharmaceutical Sciences, BioPharmaceuticals R&D, AstraZeneca, Gothenburg, Sweden. [4] Department of Molecular Cell Biology, The Norwegian Radium Hospital Montebello, N-0379 Oslo, Norway. [5] Department of Clinical Molecular Biology, University of Oslo and Akershus University Hospital, 1478 Lørenskog, Norway. [6] Department of Tumor Biology, The Norwegian Radium Hospital Montebello, N-0379 Oslo, Norway. [7] These authors contributed equally: Benan J. Mathai, Matthew Yoke Wui Ng. [8] These authors contributed equally: Laura Trachsel-Moncho, Laura R. de la Ballina. ✉email: michael.munson@astrazeneca.com; anne.simonsen@medisin.uio.no

The selective degradation of mitochondria by autophagy (mitophagy) is important for cellular homoeostasis and disease prevention. Defective clearance of damaged mitochondria is linked to the development of neurodegenerative diseases such as Parkinson's disease (PD) and has also been linked to cancer[1]. Damaged mitochondria have the potential to leak dangerous reactive oxygen species causing cell damage and ultimately death, so their rapid clearance is favoured. Mitophagy achieves this by sequestration of mitochondria into double-membrane structures termed autophagosomes that transport and deliver material to the lysosome for degradation[2]. Elucidation of the molecular mechanisms of mitophagy has largely focused upon hereditary forms of PD and the role of the genes *PINK1* and *Parkin* (*PRKN*) in mediating mitophagy in response to mitochondrial depolarisation[3]. Such studies have shown that selective recognition of damaged mitochondria involves PINK1-mediated phosphorylation of ubiquitin and PRKN, leading to further ubiquitination of outer mitochondrial membrane proteins that are recognised by specific autophagy receptors that interact with LC3 and GABARAP proteins in the autophagy membrane to facilitate mitophagosome formation. Whilst PRKN function has been linked to regulation of stress-induced mitophagy in vivo (e.g exhaustive exercise[4], alcohol-induced liver disease[5], myocardial infarction[6]), basal mitophagy in vivo appears to occur largely independent of PINK1/PRKN. This has been demonstrated across mice, fly and zebrafish models[7–9]. Consequently, further characterisation of the mechanisms involved in PRKN-independent basal mitophagy pathways is needed to understand their role in normal physiology and disease development. One such pathway involves HIF1α/hypoxia-dependent upregulation of the mitophagy receptor BNIP3 that has been particularly well characterised for its role in the clearance of red blood cell mitochondria[10]. Several small molecules have been found to stabilise HIF1α and replicate a hypoxia-induced mitophagy phenotype without the requirement for hypoxic conditions[11], including cobalt chloride, dimethyloxallyl glycine (DMOG) and iron chelators such as deferiprone (DFP)[12,13].

Whilst much progress has been made in our understanding of selective autophagy and the proteins involved, little is known about the lipids and lipid-binding proteins involved in cargo recognition and autophagosome biogenesis during selective autophagy. The formation of autophagosomes relies upon a multitude of trafficking processes to manipulate and deliver lipids to the growing structure and several proteins containing lipid interaction domains have been found to play important roles in modulating autophagy[14].

Here, we carry out an imaging-based screen to examine whether human proteins containing lipid-binding domains have unidentified roles in HIF1α-dependent mitophagy. We identify a shortlist of eleven candidates that regulate mitophagy. In particular, we show that two kinases, GAK and PRKCD, specifically regulate HIF1α-dependent mitophagy without affecting PRKN-dependent mitophagy and that these kinases also regulate basal mitophagy in vivo. Therefore, these kinases represent targets for the study and regulation of basal mitophagy.

## Results

**Induction and verification of mitophagy in U2OS cells**. To monitor mitophagy in cultured cells, U2OS cells were stably transfected to express a tandem tag mitophagy reporter containing EGFP-mCherry fused to the mitochondrial localisation sequence (MLS) of the mitochondrial matrix protein NIPSNAP1 (hereafter termed inner MLS (IMLS) cells) in a doxycycline-inducible manner[15]. U2OS cells contain low endogenous PRKN levels that are insufficient to induce mitophagy in response to

mitochondrial membrane depolarisation[16]. By contrast, treatment with the iron chelator DFP for 24 h caused movement of mitochondria to lysosomes, as a yellow mitochondrial network was seen under normal conditions, while red-only punctate structures appeared in DFP-treated cells, representing mitochondria in lysosomes (EGFP signal quenched due to the acidic pH) (Fig. 1a, b). Co-staining for the inner mitochondrial protein TIM23 verified that the EGFP-mCherry tag was localised to the mitochondrial network (Fig. 1a). As predicted, the number of red-only structures dropped drastically in cells treated with the V-ATPase inhibitor Bafilomycin A1 (BafA1) for the final 2 h of DFP treatment[17] (Fig. 1b). Similarly, siRNA-mediated depletion of the key autophagy inducer ULK1 before DFP addition significantly reduced the formation of red-only structures (Fig. 1b). IMLS reporter localisation was further validated by correlative light and electron microscopy (CLEM) following DFP treatment. Indeed, yellow network structures observed by confocal fluorescence microscopy corresponded to mitochondrial structures seen by EM, whereas red-only structures demonstrated typical autolysosome morphologies (Fig. 1c).

We could also demonstrate DFP-induced mitophagy biochemically by measuring the enzymatic activity of the mitochondrial matrix protein citrate synthase[18]. Treatment with DFP for 24 h reduced citrate synthase activity by ~40%, which was prevented by the addition of BafA1 for the final 16 h, confirming that the reduction was due to lysosomal-mediated degradation (Fig. 1d). Finally, we were able to demonstrate by proteomic analysis that the addition of DFP for 24 h decreased the abundance of multiple mitochondrial proteins (classified by gene ontology (GO) analysis) compared to control-treated cells (Fig. 1e). Comparison of different cellular organelles and compartments by GO annotation highlighted that mitochondrial proteins resident to the inner membrane, matrix and respiratory chain complex were particularly reduced in response to DFP treatment (Fig. 1f and Supplementary Table 1). Peroxisomal and lipid droplet protein abundances were also slightly reduced, whilst proteins belonging to the ER, lysosomes or cytoskeleton showed minor differences (Fig. 1f). In contrast, proteins involved in processes defined as glycolytic showed increased abundance.

Taken together, we show that DFP treatment induces a lysosomal-dependent loss of mitochondrial proteins in U2OS cells and that mitolysosomes can be robustly quantified by image analysis in U2OS IMLS cells.

**Screening for lipid-binding protein regulators of mitophagy**. To uncover the mechanisms involved in selective recognition and turnover of mitochondria, we carried out an image-based screen monitoring the formation of red-only structures in response to DFP following siRNA-mediated knockdown of 197 putative lipid-binding proteins in U2OS IMLS cells. An initial list of proteins containing established lipid interacting protein domains (FYVE, PX, PH, GRAM, C1, C2, PROPPIN, ENTH) was identified using ExPASy Prosite (see "Methods" and Supplementary Data 1). This preliminary target list was cross-examined with several U2OS proteomic datasets and restricted to proteins validated to be expressed in U2OS cells[19,20]. We included all FYVE or PX domain-containing proteins due to the relevance of phosphatidylinositol 3-phosphate (PtdIns(3)P) binding proteins in autophagy initiation[21].

The primary screen was carried out using a pool of three different siRNA oligonucleotides sequences per target. Significant changes were observed with siRNA treatment in the presence of DFP, which are plotted as fold change relative to non-targeting (siNT) samples and grouped based upon lipid-binding domains (Fig. 2a–e). As previously shown, BafA1 treatment or depletion of

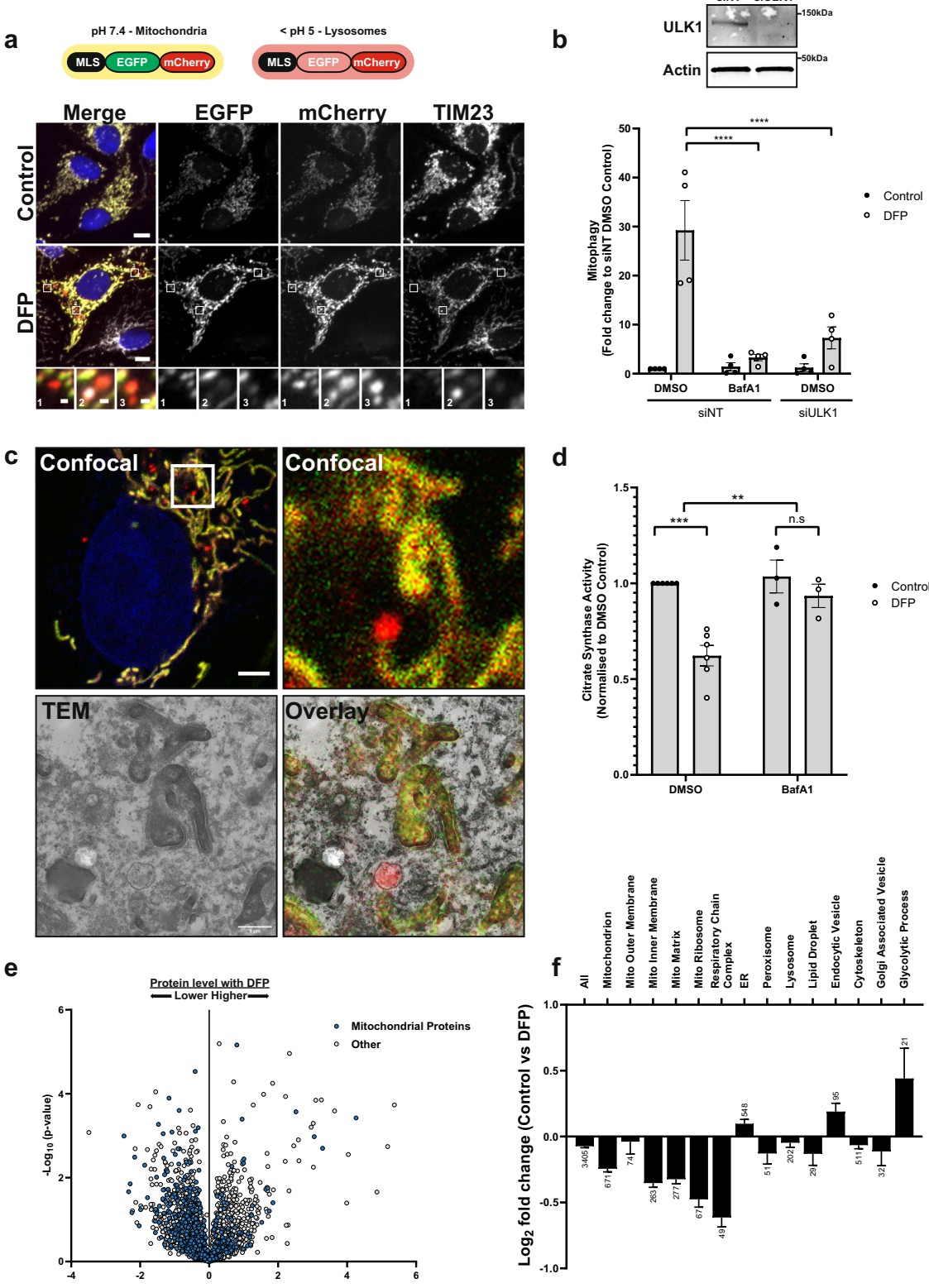

ULK1 strongly inhibited DFP-induced formation of red (mito-lysosome) structures (Figs. 1b and 2a–e, red bars). Significantly increased levels of mitophagy were also seen following knock-down of HS1BP3 (Fig. 2a), a negative regulator of starvation-induced autophagy that has not previously been examined in mitophagy[22]. It is worth noting that we found relatively few hits

from proteins containing a FYVE or PX domain compared to C1, C2 or GRAM domain-containing proteins (Fig. 2f).

**GAK and PRKCD identified as DFP mitophagy regulators by siRNA screening.** To validate prospective positive and negative regulators that demonstrated significant changes in the primary

**Fig. 1 Measuring DFP-induced mitophagy in vitro. a** U2OS cells stably expressing an internal MLS-EGFP-mCherry (IMLS) reporter that is pH-responsive (yellow at neutral, red at acidic pH) were incubated for 24 h ±1 mM DFP followed by PFA fixation, antibody staining for TIM23 (Alexa Fluor-647) and widefield microscopy. Scale bar = 10 μm, inset = 0.5 μm. **b** U2OS IMLS cells were transfected with 7.5 nM siRNA non-targeting control (siNT) or siULK1 for 48 h prior to 24 h treatment ±1 mM DFP in the presence or absence of 50 nM BafA1 for the final 2 h. Western blot from cell lysates shows representative ULK1 knockdown level. The graph represents the mean red-only area per cell from fluorescence images normalised to control DMSO siNT cells ± SEM from n = 4 independent experiments. Significance was determined by two-way ANOVA followed by Tukey's multiple comparison test. **c** U2OS IMLS cells treated with 1 mM DFP as in (**a**) and fixed for CLEM analysis. Inset of the cell area in the white box is shown by confocal analysis and EM section along with EM overlay. Scale bar = 5 μm, inset = 1 μm. **d** Citrate synthase activity from U2OS cells treated for 24 h ±1 mM DFP with final 16 h in the presence of 50 nM BafA1 or DMSO, values are normalised to DMSO control from n = 6 (DMSO) or n = 3 (+BafA1) independent experiments ± SEM. Significance was determined by two-way ANOVA followed by Sidak's multiple comparisons test. **e** U2OS whole-cell protein abundance was determined by mass spectrometry following treatment ±1 mM DFP 24 h. Mitochondrial proteins identified by GO analysis (term = mitochondrion) are highlighted in blue. **f** Mean T test difference between control and DFP samples for peptides identified in e matching GO terms related to cellular organelles. Bars represent Log2 fold change (control vs DFP) ± SEM from n = 4 independent experiments, number on bars indicate how many protein targets are included in GO analysis. **P < 0.01, ***P < 0.001, ****P < 0.0001 and n.s. = not significant in all relevant panels. For precise P values, see the source data file.

screen, we selected 29 candidates for a secondary deconvolution screen where the three siRNA oligonucleotides used in the primary screen were examined individually in U2OS IMLS cells. Their effect on DFP-induced mitophagy was analysed and quantified by high-content imaging and the level of residual target mRNA was quantified by qPCR. The mitophagy phenotype of the individual siRNA oligos was then correlated to their knockdown efficiencies and compared to the mitophagy effect of the siRNA pool used in the primary screen (Supplementary Fig. 1a). As the knockdown efficiencies for some of the targets were limited in the secondary screen, we carried out a tertiary screen for all targets that were found to significantly affect mitophagy in the primary screen, using individual siRNA oligos at a higher concentration (Supplementary Fig. 1b). Based on the results of the secondary and tertiary screen, we highlighted eleven targets where at least two of the three oligos modulated DFP-induced mitophagy (Fig. 3a).

To identify candidates with the highest possible relevance for mitophagy, we examined interacting proteins for each of these eleven candidates using network analysis of protein interaction data (see "Methods"). Interacting proteins were subjected to GO analysis and grouped based upon compartments of interest, including mitochondria, autophagy and endolysosomal compartments. The percentage of interacting proteins in each group was compared to the average percentage for all proteins screened. Of interest, AKAP13 had a high number of interactors linked to mitochondria and autophagy. In addition, the two kinases GAK and PRKCD had higher numbers of autophagy and mitochondria-linked proteins respectively than all screened proteins (Fig. 3b and Supplementary Fig. 2a, b). As several mitophagy receptor proteins involved in HIF1α-induced mitophagy are regulated by phosphorylation (including BNIP3 and BNIP3L[23,24]), we decided to further characterise the role of GAK and PRKCD in mitophagy. Further indicating a role of GAK and PRKCD in mitophagy, GAK has been linked as a risk factor for PD[25] and PRKCD has previously been noted to translocate to the mitochondria[26].

All three oligonucleotides targeting PRKCD and two of three targeting GAK caused significant inhibition of DFP-induced mitophagy in U2OS IMLS cells (Fig. 3a and Supplementary Fig. 3a). Furthermore, the level of citrate synthase activity was significantly decreased in DFP-treated cells, but not in cells depleted of PRKCD or GAK (Fig. 3c). Some loss of citrate synthase was still observed, likely due to incomplete target knockdown as noted by qPCR in the secondary screen (Supplementary Fig. 1a). By contrast to protein depletion, stable overexpression of PRKCD resulted in enhanced mitophagy (~twofold higher) in response to DFP treatment (Supplementary Fig. 3c–e).

To understand whether GAK or PRKCD localise to mitochondria upon DFP treatment, U2OS cells were enriched for mitochondria by differential centrifugation. Successful fractionation was confirmed by enrichment of the mitochondrial proteins TIM23 and COXIV (Fig. 3d). PRKCD was strongly enriched in mitochondrial fractions under both control and DFP-inducing conditions, while GAK was absent from the mitochondria fraction (Fig. 3d). Mitochondrial localisation of PRKCD was further validated by immunofluorescence staining of endogenous PRKCD in U2OS IMLS cells, showing a striking co-localisation with the IMLS reporter in both the presence and absence of DFP (Fig. 3e). Furthermore, western blot analysis indicated a partial loss of PRKCD in a DFP-dependent manner that was rescued by BafA1 treatment, similar to TIM23 (Fig. 3f, g), suggesting that PRKCD is degraded by mitophagy. Unexpectedly, PRKCD lacking C1 and C2 domains (Δlipid-binding domains—ΔLBD) was still present in the mitochondrial fraction (Fig. 3h), indicating that protein interactions may facilitate mitochondrial localisation of PRKCD, although lipid binding is important for its activation[27].

Attempts to identify endogenous GAK localisation by immunofluorescence were unsuccessful with non-specific staining observed. Expression of exogenous EGFP-GAK indicated a perinuclear localisation (Supplementary Fig. 3b) that likely reflects earlier reports of Golgi localisation[28]. We identified EGFP-GAK positive vesicles moving in the close vicinity of mitochondrial networks (Supplementary Fig. 3b).

**The kinase activity of GAK and PRKCD is required for functional mitophagy.** As GAK and PRKCD are both serine–threonine protein kinases we next sought to determine whether their kinase activities are required for DFP-induced mitophagy. We first tested two kinase inhibitors targeting GAK, IVAP1966 and IVAP1967, with a Kd of 80 and 190 nM, respectively (Supplementary Fig. 4a)[29]. Concomitant dosing of IMLS cells with DFP and these GAK inhibitors demonstrated dose-dependent inhibition of mitophagy with IVAP1967, while IVAP1966 had no effect (Supplementary Fig. 4b). We next tested a more potent and specific GAK inhibitor (SGC-GAK-1, termed GAKi here) with a Kd of 3.1 nM and IC$_{50}$ of 110 nM, which also has a negative control probe (SGC-GAK-1N, termed GAKc here, GAK IC$_{50}$ = > 50 μM) (Supplementary Fig. 4a) as well as a second control that accounts for off-target effects of GAKi (HY-19764, RIPK2 inhibitor)[30]. Using this inhibitor set, we found that DFP-induced mitophagy was significantly reduced in a dose-dependent manner with GAKi by 40% and 60% at 5 and 10 μM, respectively (Fig. 4a, b), while GAKc and RIPK2i had no effect (Fig. 4a, b), providing evidence that GAK kinase activity is required for functional DFP mitophagy. The effect of GAKi on mitophagy was

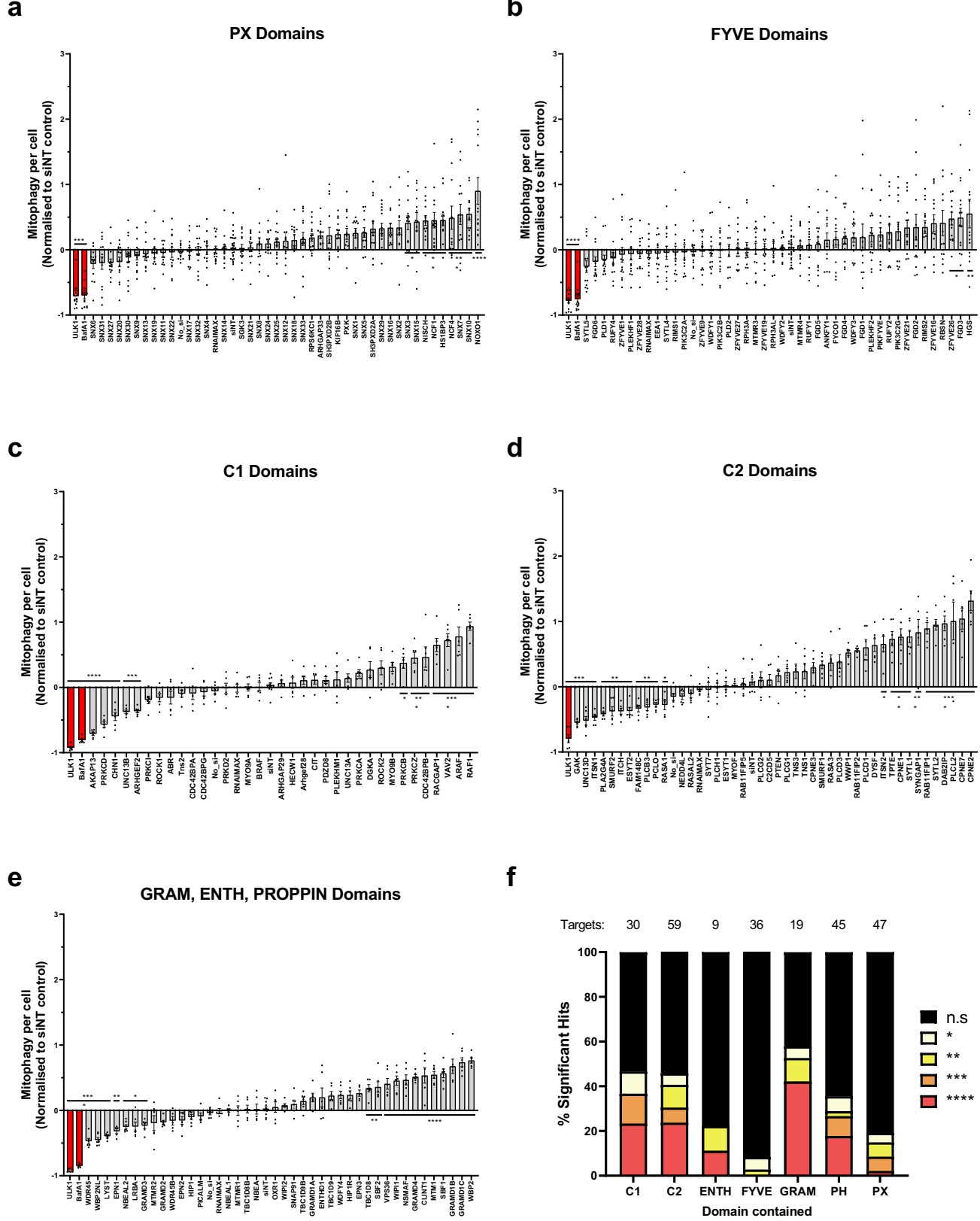

not due to reduced cell growth during DFP treatment (Supplementary Fig. 4c).

To investigate the role of the PRKCD kinase activity in mitophagy, we utilised the PKC family inhibitors (PKCi)

enzastaurin (ES) and sotrastaurin (SS), neither of which is solely specific for PKC Delta due to isozyme similarity within the PKC family. Both compounds strongly inhibited DFP-induced mitophagy in a dose-dependent manner (Fig. 4a, b) with enzastaurin

**Fig. 2 siRNA screen for lipid-binding proteins involved in DFP-induced mitophagy. a–e** Primary siRNA screen data. U2OS cells were transfected with a pool of three sequence variable siRNA oligonucleotides per gene target (2.5 nM per oligo) for 48 h before the addition of 1 mM DFP for 24 h. Cells were PFA fixed and imaged using a ×20 objective (35 fields of view per well). The red area per cell was normalised to the average of siNT controls and adjusted so that the DFP siNT control was 0 from $n = 14$ plates (**a**, **b**) or $n = 6$ plates (**c–e**) ±SEM. siULK1 and BafA1 (red bars) are positive controls. siRNA targets containing similar lipid-binding domains were assayed and plotted together as shown for **a** PX domains, **b** FYVE domains, **c** C1 domains, **d** C2 domains, **e** GRAM, ENTH, PROPPIN domains. **f** Summary of the significance of different lipid-binding domains relative to the total tested from (**a–e**) proteins containing more than one type of domain are represented in each category. Significance was determined by one-way ANOVA followed by Dunnett's multiple comparison test to the siNT control where $*P < 0.05$, $**P < 0.01$, $***P < 0.001$, $****P < 0.0001$ and n.s. = not significant in all relevant panels. For precise $P$ values, see the source data file.

slightly more potent than sotrastaurin. The conventional (PRKCA, PRKCB), novel (PRKCD) and atypical (PRKCZ) PKC family members were included in the primary siRNA screen, with only siPRKCD showing significantly reduced DFP-induced mitophagy of −57% (Fig. 2c). To further investigate the potential effects of other PKC isozymes, all PKC isoforms were depleted in the U2OS IMLS cells, which demonstrated that multiple novel PKC isoforms decreased DFP-induced mitophagy, with PRKCD showing the strongest effect (Supplementary Fig. 4d). Novel PKCs do not require calcium, but diacylglycerol (DAG) for regulation of their activity, suggesting DAG may be important for DFP-induced mitophagy[27]. The use of pan-PKC kinase inhibitors may therefore be beneficial for exploring the role of PKC family members in DFP-induced mitophagy due to potential isozyme redundancy.

To further confirm a role for GAK and PKC kinase activities in mitophagy, we determined the level of citrate synthase activity following treatment with GAKi, GAKc and PKCi. As seen in Fig. 4c, GAKi and PKCi strongly blocked DFP-induced loss of citrate synthase activity whilst this was not seen with DMSO or GAKc control samples. Moreover, the addition of GAKi for 24 h decreased the DFP-induced loss of inner mitochondrial membrane proteins (MTCO2, COXIV, TIM23), matrix proteins (PDH, NIPSNAP1) and outer membrane protein TOM20 as examined by western blot analysis (Fig. 4d). By contrast, loss of the outer mitochondrial protein FUNDC1 was not affected (Fig. 4d). This is perhaps not surprising as some outer mitochondrial proteins, such as MFN2, are known to undergo proteasomal degradation upon induction of DFP-induced mitophagy[12]. We further confirmed the effect of GAK on the loss of mitochondrial proteins by mass spectrometry analysis. DFP treatment with DMSO or GAKc induced a loss of mitochondrial proteins across multiple mitochondrial compartments (determined by GO Analysis), but this was blocked by GAKi treatment, indicating that mitochondrial turnover is strongly impaired when GAK kinase activity is blocked (Fig. 4e). The loss of mitochondrial proteins was more variable with PKCi after 24 h (Fig. 4d and Supplementary Fig. 4e), however, treatment of up to 48 h DFP + PKCi resulted in robust inhibition of TIM23, COXIV and MTCO2 protein degradation (Fig. 4f).

**GAK and PRKCD do not regulate the HIF1α pathway**. To understand how depletion or inhibition of GAK or PKCs prevents DFP-induced mitophagy, we examined whether treatment with GAKi or PKCi affected the HIF1α pathway. HIF1α stabilisation and expression of the HIF1α-responsive genes BNIP3 and BNIP3L (NIX) are important for driving mitophagy triggered by iron chelation with DFP[12,31]. As expected, BNIP3/3L mRNA and protein levels were induced upon DFP treatment with no change in HIF1α mRNA levels (Fig. 4d and Supplementary Fig. 5a, b), but were not significantly affected by treatment with GAKi, suggesting that GAK does not regulate mitophagy through the HIF1α response. While the PKCi sotrastaurin had no significant effect on BNIP3/3L expression, enzastaurin somewhat reduced the BNIP3L transcript level (Supplementary Fig. 5a), which may explain the relatively

higher potency of enzastaurin-blocking mitophagy (Fig. 4b). We focused on sotrastaurin for further PKCi experiments as our other data indicated there was an important role for the kinase activity beyond the decrease in BNIP3L seen with enzastaurin.

BNIP3/3L function as autophagy receptors during HIF1α-induced mitophagy by recruiting ATG8 proteins through specific LC3 interacting regions (LIRs) and phosphorylation promote their function in mitophagy[32,33]. To check whether BNIP3/3L are targets for GAK or PRKCD-mediated phosphorylation, cell lysates from U2OS cells treated or not with DFP together with GAKi or PKCi were analysed using phos-tag acrylamide gels that significantly retard the migration of phosphorylated proteins[34]. Importantly, neither the protein expression of BNIP3/3L with DFP nor their migration patterns were changed in GAKi, GAKc and PKCi-treated samples (Supplementary Fig. 5b), suggesting that neither BNIP3 nor BNIP3L are targets for direct phosphorylation by PRKCD or GAK. The uppermost band visible by phos-tag blot was sensitive to treatment with calf-intestinal phosphatase (CIP), confirming these represent P-BNIP3 and P-BNIP3L species (Supplementary Fig. 5b).

**GAK and PRKCD kinase activities do not regulate PRKN-dependent mitophagy or starvation-induced autophagy**. As both GAK and PRKCD regulate DFP-induced mitophagy without affecting the HIF1α pathway, we next sought to examine whether these kinases also regulated PRKN-dependent mitophagy. The U2OS IMLS-EGFP-mCherry cell line was transduced with lentivirus to constitutively overexpress PRKN (untagged), to permit induction of mitophagy in response to mitochondrial depolarisation such as that induced by the $H^+$ ionophore CCCP[35,36]. Indeed, treatment with CCCP for 16 h led to significant induction of mitophagy and near-total loss of the mitochondrial network as seen by accumulation of red-only structures (Fig. 5a), reduced citrate synthase activity (Fig. 5b) as well as the loss of the mitochondrial matrix protein PDH, along with PRKCD that also localises on mitochondria (Fig. 5c, d). Importantly, we could block PRKN-dependent CCCP-induced mitophagy by co-treatment with the ULK1/2 kinase inhibitor MRT68921 or the lysosomal inhibitor BafA1 (Fig. 5a–d)[17,37]. Co-treatment with GAKi, GAKc or PKCi, however, had no effect (Fig. 5a–d), suggesting that GAK and PRKCD kinase activities are dispensable for PRKN-mediated mitophagy under these conditions. We next considered whether GAKi or PKCi could impair starvation-induced autophagy. Cells were incubated in nutrient starvation media (EBSS) for 2 h and the autophagic flux examined by immunostaining for endogenous LC3B in the absence or presence of BafA1, as analysed by fluorescence microscopy (Fig. 5e, f) or immunoblotting (Fig. 5g). Whilst incubation in EBSS increased LC3 flux relative to the control, the addition of GAKi or GAKc had no effect on the starvation-induced autophagic flux. The addition of PKCi did however lead to slightly elevated LC3-II levels in starved cells in the absence of BafA1 (Fig. 5e–g).

Taken together, we show that while GAK and PRKCD kinase activities are required for efficient DFP-induced mitophagy, they

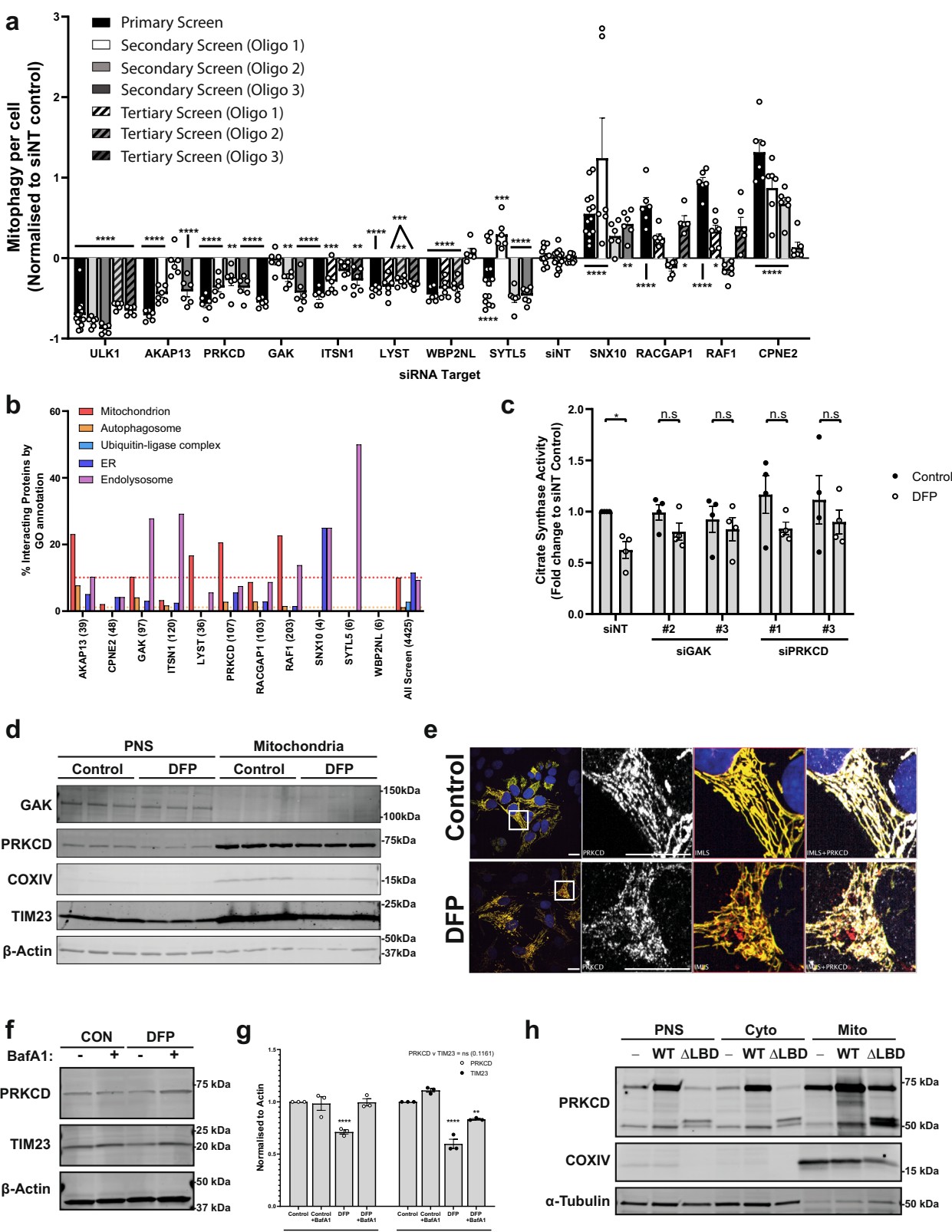

are dispensable for PRKN-dependent mitophagy and starvation-induced autophagy.

**PRKCD inhibitors reduce ULK1 initiation complex assembly.** To further elucidate how GAK or PKC kinase inhibition may block DFP-induced mitophagy, we examined the recruitment of

core autophagy machinery components to mitochondria. U2OS IMLS cells treated with DMSO or DFP were immunostained for endogenous ATG13, LC3B, ULK1 or WIPI2 and examined by confocal microscopy. In many cases, the early autophagosome structures induced by DFP treatment were localised at or close to mitochondria, likely representing mitophagosome start sites

**Fig. 3 GAK and PRKCD are regulators of DFP-induced mitophagy. a** Summary of significant targets identified across primary, secondary (7.5 nM individual siRNA oligos) and tertiary screens (15 nM each oligo) siRNA screens. Cells were transfected for 48 h prior to 24 h of 1 mM DFP treatment. Bars represent mean fold change in mitophagy relative to the siNT controls ± SEM from n = 12 (siNT, primary + secondary), n = 13 (SNX10, primary), n = 14 (SYTL5 + ULK1, primary), n = 17 (siNT, secondary) or n = 6 (all others) independent plates. Significance was determined by one-way ANOVA followed by Dunnett's multiple comparison test to the siNT control. **b** Protein–protein interaction networks for candidate proteins (see "Methods") were plotted by % of interacting proteins belonging to each highlighted compartment, value in brackets represents total number of interacting proteins. Dashed lines indicate average values from all screened proteins for mitochondria (red) or autophagosome (orange). **c** siRNA treatment with indicated oligos for 48 h prior to 24 h treatment ± 1 mM DFP and subsequent analysis of citrate synthase activity levels. Values were normalised to the siNT control and plotted ± SEM for n = 4 independent experiments. Significance was determined by two-way ANOVA followed by Sidak's multiple comparison test. **d** U2OS cells treated ± 1 mM DFP for 24 h were enriched from a post-nuclear supernatant (PNS) for mitochondria followed by western blotting for the indicated proteins. **e** U2OS IMLS cells treated ± 1 mM DFP for 24 h followed by PFA fixation and staining for endogenous PRKCD (Alexa Fluor-647). Scale bar = 20 μm. **f** U2OS cells treated ± 1 mM DFP for 24 h ± 50 nM BafA1 for the final 16 h and blotted for the indicated proteins. **g** Quantitation of PRKCD and TIM23 levels to β-actin in (**f**) from n = 3 independent experiments ± SEM. Significance was determined by two-way ANOVA followed by Dunnett's multiple comparison test to the control. **h** U2OS cells (−) or those stably expressing PRKCD wild type (WT) or PRKCD ΔC1 ΔC2 (Δlipid-binding domains—LBD) were enriched from a post-nuclear supernatant (PNS) for cytosol and mitochondrial fractions followed by western blotting for the indicated proteins. Representative from n = 2 experiments *P < 0.05, **P < 0.01, ***P < 0.001, ****P < 0.0001 and n.s. = not significant in all relevant panels. For precise P values, see the source data file.

(Fig. 6a–d). Using high-content imaging we found that the formation of WIPI2, ATG13 or ULK1 puncta were all strongly induced (~fivefold) in response to 24 h DFP treatment (Fig. 6e, f and Supplementary Fig. 6a–d). Treatment with GAKi or GAKc did not affect the number of WIPI2, ATG13 or ULK1 puncta observed following 24 h of DFP treatment (Fig. 6e, f and Supplementary Fig. 6a–d). Of note, control cells treated with GAKi only (no DFP) displayed an increased number of WIPI2 puncta (Supplementary Fig. 6a, d), but the reason for this is not apparent. Examination of LC3 puncta co-localisation with mitochondria did, however, reveal a decrease upon GAKi treatment, suggesting mitochondrial loading into autophagosomes may be affected (Fig. 6g). Co-treatment of cells with DFP and PKCi caused a dose-responsive decrease in the number and size of ULK1 and ATG13 puncta (and to a lesser extent WIPI2) (Fig. 6e, f and Supplementary Fig. 6d). By contrast, overexpression of PRKCD increased ULK1 puncta formation in response to DFP treatment (Supplementary Fig. 6e).

Whilst ULK1 complex localisation appears to be affected by PKCi, neither the phosphorylation of ATG13 at S318 (a known ULK1 site)[38] nor the phosphorylation of AMPK at T172 (an activation site)[39] that occur in response to DFP treatment was impaired or altered by co-treatment with PKCi or GAKi (Fig. 4d). Similarly, DFP treatment alone did not inhibit phosphorylation of mTORC1 substrates (ULK1 P-757, p70S6K P-389) like starvation-induced autophagy, suggesting DFP-induced mitophagy is mTORC1 independent (Supplementary Fig. 6f). However, GAKi treatment in combination with DFP blocked phosphorylation of mTORC1 substrates p70S6K and ULK1, though it is unclear if this inhibition may interfere with the standard DFP signalling response (Fig. 6f).

We thus conclude that the kinase activity of PRKCD is required for successful assembly and formation of the ULK1 complex during DFP-induced mitophagy, but that it does not affect ULK1 activity directly. As PRKCD is localised to mitochondria, a failure to generate an initiation structure upon its depletion or inactivation likely explains the reduced level of mitophagy observed.

**GAK inhibition alters mitochondrial and lysosomal morphology.** As GAK inhibition did not affect HIF1α signalling, ULK1 activity or ATG13/ULK1 recruitment, but appeared to regulate mitochondrial loading, we next examined mitochondrial morphology. IMLS cells co-treated with DFP and siGAK (Supplementary Fig. 3a) or GAKi (Fig. 4a) demonstrated abnormal mitochondrial network structures compared to control cells. Using live-cell

microscopy, we observed a collapsed mitochondrial network with clumped mitochondrial regions and reduced red structures in IMLS cells co-treated with DFP and GAKi that were not seen with GAKc (Supplementary Movie 1). Confocal microscopy confirmed this and by contrast, cells co-treated with DFP and PKCi retained an elongated network (Fig. 7a). Cells treated with a combination of Oligomycin and Antimycin A (O + A) or CCCP, known to depolarise mitochondria, caused fragmented network phenotypes (Fig. 7a) without induction of mitophagy as this was carried out in IMLS cells without PRKN overexpression. To quantify mitochondria morphology, images of U2OS cells depleted of the fission enzyme DRP1 or the fusion enzyme OPA1 (causing tubulated or fragmented mitochondrial phenotypes respectively, Supplementary Fig. 7a), were applied to train a cell classifier in CellProfiler Analyst. Applying this, we confirmed that GAKi-treated cells exhibit a hyperfused mitochondria phenotype while no morphological differences were detected in GAKc or PKCi-treated samples (Fig. 7b). The observed GAKi phenotype was not caused by a general defect in the fission/ fusion ability of the mitochondrial network, as mitochondrial fragmentation could still be induced by co-treatment of GAKi with CCCP and DFP (Fig. 7c, d). Importantly, GAKi still prevented the formation of red-only structures despite CCCP treatment, indicating that GAK is important for the proper uptake of fragmented mitochondria into autophagosomes.

We next examined GAKi-treated U2OS IMLS cells by CLEM. This demonstrated condensed parallel layers of mitochondria that were not fused but rather stacked closely with one another (Fig. 7e). In addition, large autolysosome structures were observed, which may indicate lysosomal defects following GAKi treatment (Fig. 7e). Indeed, an increase in the number of lysosomal structures was detected in cells stained for the late endosome/lysosome marker LAMP1 following treatment with GAKi (Fig. 7f, g). As GAKi did not inhibit PRKN-dependent mitophagy (Fig. 5a–d) or starvation-induced autophagy (Fig. 5e–g), we reasoned that the large autolysosomes seen in GAKi-treated cells retain their acidity and degradative capacity. In line with this, lysotracker-positive structures were detected in cells treated with GAKi or GAKc in the presence or absence of DFP (Supplementary Fig. 7b). Supporting a role for GAK in the regulation of lysosome size, we found that nuclear localisation of TFEB-mCherry, a transcription factor for lysosomal biogenesis, was increased in GAKi-treated cells, both in the absence and presence of DFP (Supplementary Fig. 7c, d), potentially explaining the increased lysosomal abundance observed (Fig. 7f, g). It is interesting to note that GAKi also inhibits mTORC1 activity in DFP-treated cells (Supplementary Fig. 6f), similarly to

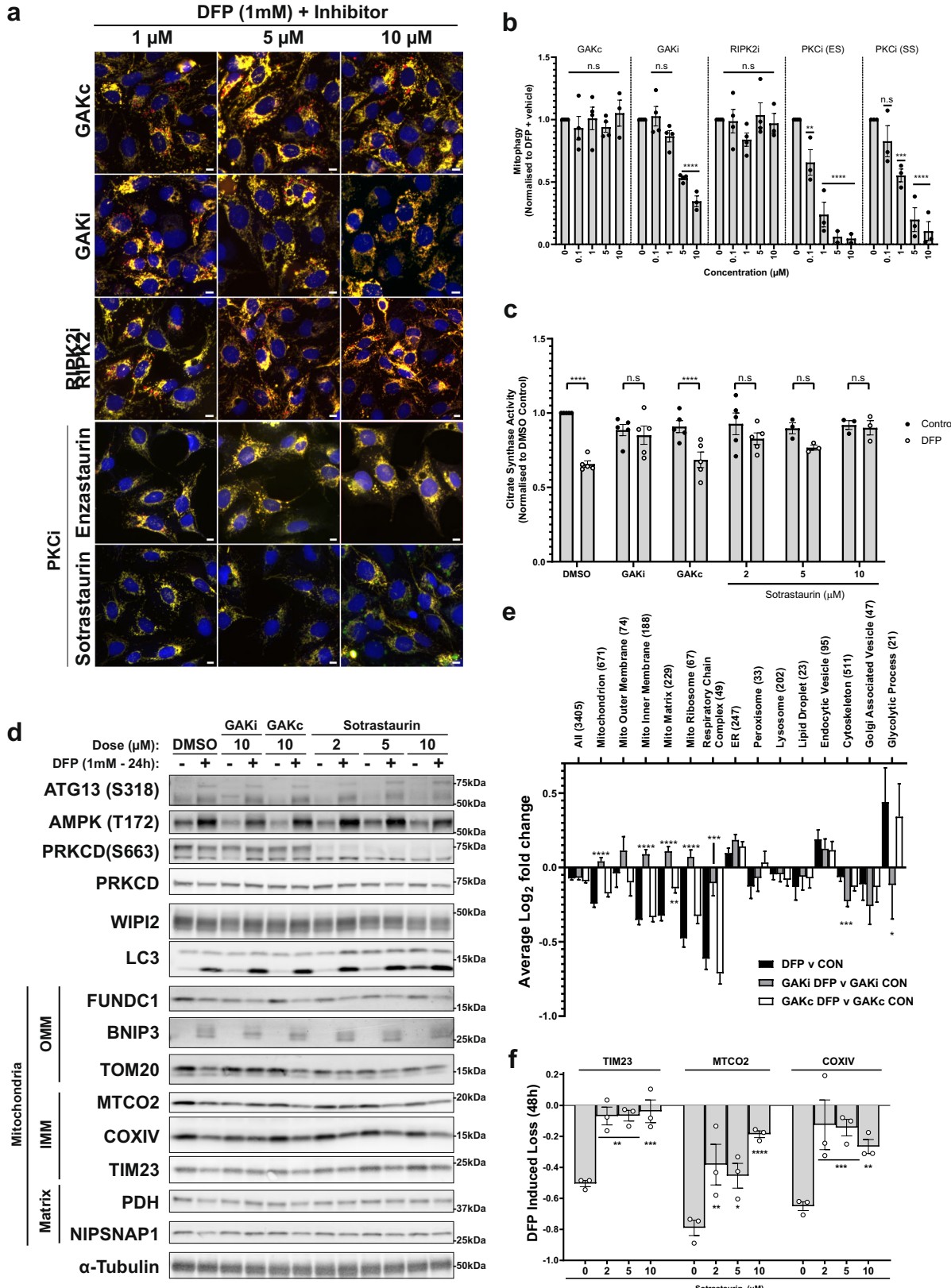

the mTORC1 inhibitor Torin1 that also stimulated nuclear localisation of TFEB-mCherry (Supplementary Fig. 7c, d).

Thus, we believe that the mitophagy defect induced by GAKi is partially due to inefficient cargo loading or delivery to lysosomes rather than a lysosomal defect, despite apparent lysosomal abnormalities present.

**Mass spectrometry with GAKi.** To try to identify relevant substrates for GAK kinase-dependent regulation of DFP-induced mitophagy, we carried out the phosphoproteomic analysis of cells treated with DFP in combination with GAKi or GAKc. GAK is currently defined as an understudied kinase[40] and GAKi was developed by the SGC consortium to further understanding

**Fig. 4 GAK and PRKCD kinase activity regulate DFP-induced mitophagy. a** Fluorescence images of U2OS IMLS cells treated with 1 mM DFP for 24 h in the presence of indicated inhibitors at stated concentrations (1–10 μM). Scale bar = 10 μm **b** Quantitation of cells treated as in (**a**) for mean red-only structures normalised to DMSO + DFP control treatment ± SEM. Significance was determined by two-way ANOVA followed by Dunnett's multiple comparison test to the DMSO + DFP control from a minimum of $n = 3$ independent experiments. **c** Citrate synthase activity of U2OS cells treated 24 h ± 1 mM DFP in combination with 10 μM GAKi, GAKc or 2–10 μM sotrastaurin. Values represent mean citrate synthase activity normalised to DMSO control and plotted ± SEM from $n = 3$ (sotrastaurin 5/10 μm) or $n = 5$ independent experiments. Significance was determined by two-way ANOVA followed by Sidak's multiple comparison test. **d** U2OS cells were treated ± 1 mM DFP 24 h with GAKi or GAKc (10 μM), sotrastaurin (2–10 μM) or DMSO control and western blotted for indicated proteins, including outer mitochondrial membrane (OMM), inner mitochondrial membrane (IMM) or Matrix proteins. **e** Cellular protein abundance was determined by mass spectrometry. U2OS cells were treated ± 1 mM DFP for 24 h in addition to DMSO, GAKi or GAKc (both 10 μM) and analysed by mass spectrometry. Comparison of mean abundance of GO-annotated proteins between control and DFP-treated samples for each GAKi, GAKc and DMSO control are shown ± SEM from $n = 4$ independent experiments, value in brackets represent the number of proteins classified in the group by GO analysis. Significance was determined by two-way ANOVA followed by Dunnett's post-test to the DFP v CON sample. **f** Cells were treated as in (**d**) for 48 h and quantified by western blot by normalisation of each target to loading control. The graph represents mean DFP-induced loss of indicated proteins ± SEM from $n = 3$ independent experiments. Significance is denoted in figure where: *$P < 0.05$, **$P < 0.01$, ***$P < 0.001$, ****$P < 0.0001$ and n.s. = not significant in all relevant panels. For precise $P$ values, see the source data file.

of GAK cellular function[30]. Analysis of both the protein abundance and phospho-sites of interest (Supplementary Fig. 8) demonstrated that DFP treatment increased the abundance of several proteins in a GAKi-independent manner, including HIF1α, BNIP3L, Hexokinase I/II, LAMP1/2 and LC3B. In contrast, the abundance of SQSTM1 decreased in response to DFP, even with GAKi, further showing that lysosomes are functional in this state. Intriguingly, DFP treatment caused phosphorylation of RAB7A at S72, which is the same modification recently reported to facilitate a key step in PRKN-dependent mitophagy[41], suggesting mechanistic similarities between the two mitophagy pathways.

Increased levels of PGK1 S203, PGM1 S117 and PGM2 S165 (established phosphorylation sites to induce glycolysis) were detected upon GAKi treatment without DFP, suggesting that GAK functions as a negative regulator of glycolysis. We also see moderate changes to phosphorylation of different components of the mitochondrial fission–fusion machinery (DRP1, MFF, MFN1/2) compared to DMSO or GAKc treatment that may together explain some of the abnormal mitochondrial network phenotypes observed following GAKi treatment.

**GAK and PRKCD modulate mitophagy in vivo.** Knockout of GAK orthologues have been shown to cause embryonic lethality in *M. musculus*, *C. elegans* (*dnj-25*) and *D. melanogaster* (*dAux*). This is thought to be due to defective clathrin-dependent endocytosis, as uncoating of clathrin-coated vesicles is mediated by the J-domain of GAK[42–44]. Indeed, expression of the J-domain alone is able to rescue survival in both mice and drosophila GAK knockout models[44,45]. *C. elegans* contains two orthologues of GAK, comprising the kinase domain (gakh-1, F46G11.3, 40% overall homology) or the J-domain (dnj-25, W07A8.3, 49% overall homology) (Fig. 8a), where targeting of the latter by RNAi causes lethality during larval development[43]. As our data indicate a requirement of GAK kinase activity for efficient mitophagy in mammalian cells, we examined the effect of targeting *gakh-1* upon basal mitophagy in a *C. elegans* reporter line expressing mtRosella GFP-DsRed in body-wall muscle cells, following the concept of the IMLS reporter cell lines. *C. elegans* fed *gakh-1* RNAi demonstrated a significant decrease in the ratio of GFP to DsRed and more mitophagy events (DSRed only structures, Fig. 8b, c) compared to the RNAi control, confirming that GAK kinase activity is important for basal mitophagy in vivo.

We next sought to examine the role of PRKCD in mitophagy in vivo and targeted Prkcd in our recently reported transgenic mitophagy reporter zebrafish line, expressing zebrafish Cox8 fused to EGFP-mCherry[46]. Zebrafish contain PRKCD paralogues (prkcda and prkcdb) that are ~80% similar to one another and

~76% homologous to human PRKCD, with all major domains being highly conserved (Fig. 9a). We examined the spatio-temporal expression pattern of *prkcda* and *prkcdb* during zebrafish development up until 5 days post fertilisation (dpf). Whilst a high level of maternal *prkcda* mRNA was observed at 2 h post fertilisation (hpf), possibly indicating a role in early embryonic signalling events, both genes were expressed at similar levels throughout development (Supplementary Fig. 9a). The spatial distribution of *prkcda* and *prkcdb* were analysed by whole-mount in situ hybridisation (WM-ISH) at 5 dpf compared to a sense probe negative control and showed strong staining in the corpus cerebelli region of the hindbrain for both genes (Fig. 9b and Supplementary Fig. 9b), this is in agreement with ISH data deposited in the ZFIN database (http://zfin.org). *prkcda* expression was also detected in the eyes and the olfactory bulbs, whereas *prkcdb* was found in patches of the retina, optical tectum and the spinal cord (Fig. 9b and Supplementary Fig. 9b). We carried out immunofluorescent staining for prkcd protein and identified high levels of prkcd in the hindbrain region of zebrafish (Fig. 9c) that similarly matched the location of the hindbrain marker chondroitin sulphate (Fig. 9d)[47–49]. Thus, in later analysis, we define the hindbrain as the region marked by chondroitin sulphate (within the two dashed white lines, Fig. 9d).

We also analysed the spatio-temporal pattern of *gak* in zebrafish and observed consistent expression throughout development (Supplementary Fig. 9c), however, its spatial expression pattern varies across the development with staining of the caudal hindbrain and retina at 2 dpf, the optical tectum, neurocranium, retina and kidney at 3 dpf, the kidney and neurocranium at 4 dpf and the liver, olfactory bulb, optical tectum and retina at 5 dpf of the wild-type zebrafish larvae, with no staining of the control probe (Supplementary Fig. 9d).

To investigate a possible role for Prkcd in the mitophagy reporter line, we employed CRISPR/Cas9-mediated genome editing in zebrafish embryos using guide sequences targeting *prkcda* and *prkcdb* individually or together (Fig. 9e and Supplementary Fig. 9e), with the latter referred to as *prkcd_ab* double knockout (DKO). We verified loss of Prkcd protein levels in DKO embryos by western blot (Fig. 9e) and therefore used these embryos for in vivo mitophagy analysis.

To induce mitophagy, zebrafish larvae at 2 dpf were incubated with varying concentrations of DFP and DMOG for 24 h. DFP failed to induce mitophagy at all concentrations, however, DMOG-induced mitophagy as shown by reduced levels of Tim23 (Supplementary Fig. 9f, g), in line with a recent report[9]. We used 100 μM DMOG for further experiments, as significant lethality and morphological defects were present at higher (250 μM) concentrations (Supplementary Fig. 9h). As *prkcd_a/b*

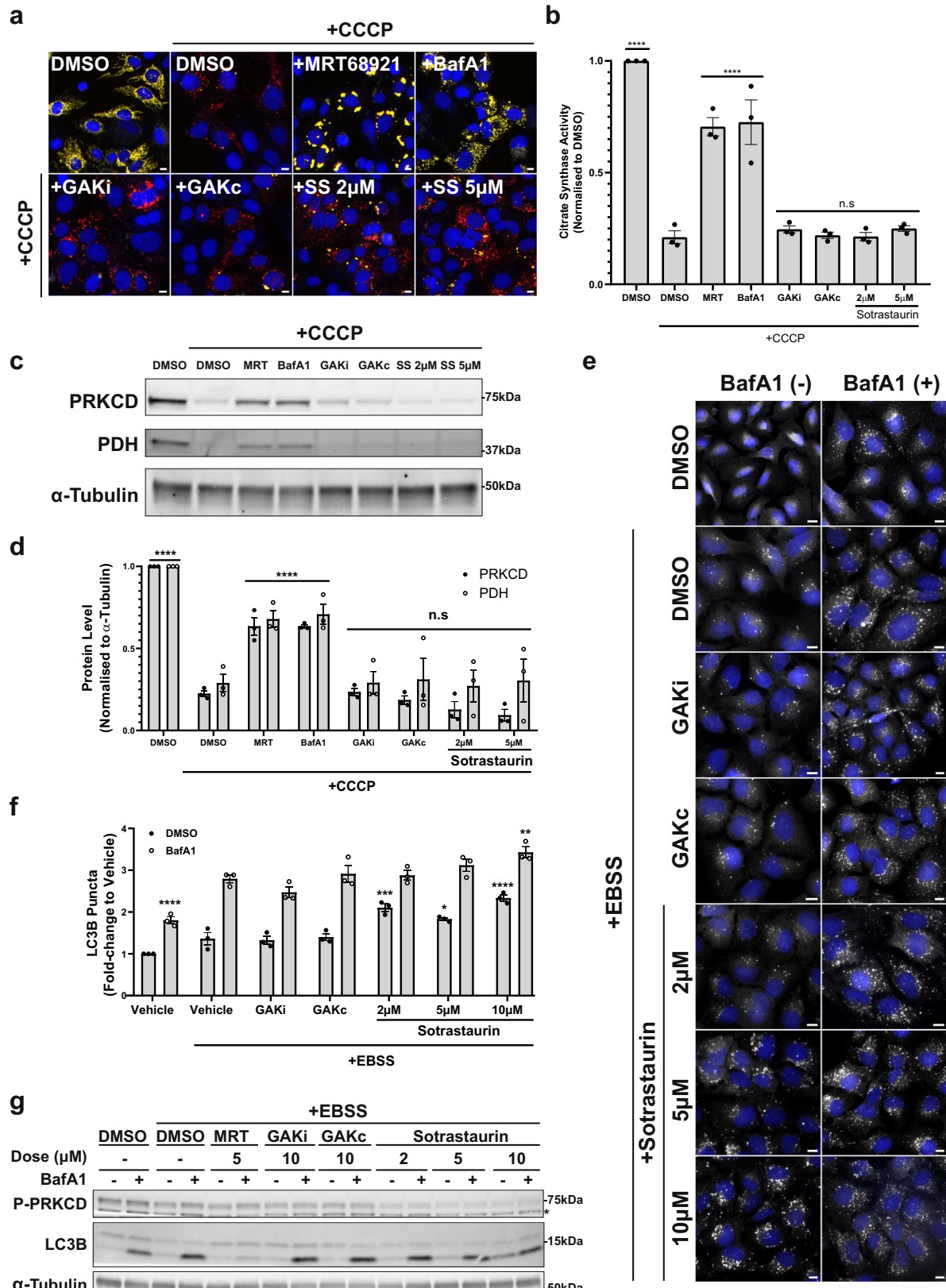

express highly in the hindbrain (Fig. 9c), we examined mitophagy in this region of DKO larvae compared to control larvae, both at basal (DMSO) and DMOG-treated conditions. Sections of fixed larva were imaged by confocal microscopy and the number of red puncta in the hindbrain region quantified. The number of red-only puncta was significantly reduced in the hindbrain of DKO

larvae under both basal and DMOG-induced conditions (Fig. 9f, g), showing an important role of Prkcd_a/b in regulating mitophagy.

We observed an inconsistent and varied movement pattern of the DKO larvae and therefore quantified their locomotor activity. Tracking and quantification of locomotion were performed in

**Fig. 5 GAK and PRKCD kinase activity are dispensable for PRKN-dependent mitophagy and starvation-induced autophagy. a** Representative fluorescence images of U2OS IMLS-PRKN cells treated for 16 h ± 20 μM CCCP and including 10 μM QVD-OPh to promote cell survival in addition to either MRT68921 (5 μM), BafA1 (50 nM), GAKi (10 μM), GAKc (10 μM) or sotrastaurin (2–5 μM). Scale bar = 10 μm. **b** Cells treated as in a and assayed for citrate synthase activity and normalised to DMSO control. Mean value plotted ± SEM from $n = 3$ independent experiments, and significance determined by one-way ANOVA followed by Dunnett's multiple comparisons to the CCCP + DMSO control. **c** Representative example of western blots from cells treated as in a and blotted for indicated proteins. **d** Quantitation of PRKCD and PDH levels normalised to from western blots in (**c**) from $n = 3$ independent experiments ± SEM. Values represent protein level normalised first to α-tubulin and subsequently normalised to the DMSO control. Significance was determined by two-way ANOVA followed by Dunnett's multiple comparison test to DMSO + CCCP treatment. **e** Representative ×20 immunofluorescence images of U2OS cells stained for endogenous LC3B and nuclei (DAPI, blue). Cells were grown in complete media or EBSS (starvation) media for 2 h with the addition of GAKi (10 μM), GAKc (10 μM) or sotrastaurin (2–10 μM) ± 50 nM BafA1, scale bar = 10 μm. **f** Quantitation of LC3B puncta from (**e**). The average LC3 puncta per cell were normalised to that of the complete media control and represent the mean ± SEM from $n = 3$ independent experiments. Significance was determined by two-way ANOVA followed by Dunnett's multiple comparison test to the EBSS vehicle-treated sample. **g** Representative western blot of cells treated as in (**e**) from $n = 3$ experiments and blotted for indicated proteins where *P-PRKCQ cross-reaction. $*P < 0.05$, $**P < 0.01$, $***P < 0.001$ and $****P < 0.0001$ and n.s. = not significant in all relevant panels. For precise $P$ values, see the source data file.

alternating light and dark conditions with a reduced swimming trend in the dark phase for both *prkcd* single KO and DKO larvae compared to the WT control (Fig. 9h). It is likely that this may be related to a hindbrain clustered motor-neuron dependent phenotype.

In conclusion, we show that the activities of both GAK and PRKCD are important for the regulation of basal mitophagy in vivo, highlighting the evolutionary relevance of these kinases in mitophagy.

## Discussion

In this manuscript, we have tested a panel of putative lipid-binding proteins for their ability to regulate DFP-induced mitophagy. We have identified 11 candidate proteins that demonstrate significant modulation of DFP-induced mitophagy and explored two of these, GAK and PRKCD, in further detail. In both cases, we show that functional kinase activity is required for their positive regulation of DFP-induced mitophagy and we have identified putative mechanisms for how each functions in mitophagy regulation. Importantly, neither GAK nor PRKCD are required for PRKN-dependent mitophagy or starvation-induced autophagy, offering targets for the specific modulation of PRKN-independent mitophagy. Critically, we find that both of these targets identified in vitro are also relevant targets in vivo for the regulation of basal mitophagy in *C. elegans* and *D. rario*, demonstrating their conserved function in mitophagy.

The linkage of GAK to mitophagy is of particular relevance to neurodegenerative disease as SNPs in *GAK* have previously been identified as a risk factor for familial PD[25] and expression changes in GAK are observed in the substantia nigra of PD patients[50]. Knockout of the drosophila homologue of GAK (Auxilin) also demonstrates Parkinsonian like mobility defects and loss of dopaminergic neurons[51]. Further work has shown that the commonly mutated PD gene, *LRRK2*, can phosphorylate Auxilin[52]. Studies of GAK have to date, however, been hindered by the limited availability of chemical tools alongside the inherent difficulty of modulating embryonic lethal genes to dissect their function. *C. elegans* presents a unique opportunity for the in vivo study of GAK kinase function due to the presence of two orthologues of human GAK, with the homologous kinase domain within a separate protein (gakh-1) than that of the developmentally essential J-domain (dnj-25)[43]. By knockdown of *gakh-1*, we were able to see >50% reduction in basal mitophagy in muscle-cell wall highlighting the importance of GAK for mitophagy. Further investigation is required to mechanistically elucidate how this functional effect is mediated, as Hif1α induction and subsequent recruitment of ULK1/ATG13 to mitophagosomes in response to DFP stimulation appears normal in cells with GAK kinase activity inhibited. However, we observed a significant

disruption of normal mitochondrial network morphology and reduced LC3-Mitochondria co-localisation in GAK deficient cells, which may reflect reduced formation or loading of mitophagosomes. Artificially inducing mitochondrial fragmentation with CCCP, was not sufficient to rescue mitochondrial degradation in GAKi-treated cells. Moreover, GAK inhibition led to the formation of enlarged lysosomes, which still seem to possess degradative potential as starvation-induced autophagy and PRKN-dependent mitophagy was largely unaffected. Mass spectrometry analysis indicates that several glycolytic enzymes are activated in response to GAKi treatment, which may indicate that more fundamental metabolic changes are occurring that alter the cellular degradation of the mitochondria.

PRKCD is a member of the large PKC kinase family and belongs to the subgroup of novel PKCs (nPKCs) including PRKCE, PRKCH and PRKCQ that are all activated independently of $Ca^{2+}$ by DAG[27]. Whilst we have primarily focused upon PRKCD due to its prominent mitochondrial localisation, it is important to note that all nPKCs reduced DFP-induced mitophagy to varying extents, suggesting that DAG and PKCs play a prominent role in the regulation of mitophagy. Usage of a pan-PKC inhibitor consistently gave a stronger and more robust inhibition of DFP-induced mitophagy compared to single PKC siRNA treatments, indicating that several isoforms could play a role or have redundant functions. Treatment with the PKC inhibitor sotrastaurin led to a significant inhibition in the recruitment of the early autophagy markers ATG13 and ULK1, thereby reducing mitophagosome formation.

Formation of mitochondrial DAG has previously been observed during oxidative stress induced by $H_2O_2$, which resulted in the recruitment of Protein Kinase D1 (PKD1) to mitochondria[53]. Recent data in mice have also implicated PKD1 in the regulation of mitochondrial depolarisation and as PKD1 is regulated by PRKCD in the activation loop at S738/S742, it is tempting to speculate that PKD1 might be involved in DFP-induced mitophagy[54]. It is also intriguing that PKD1 can bind to AKAP13, another identified regulator of mitophagy in our screen[55]. The formation of mitochondrial DAG could therefore be a key step in the regulation of PRKN-independent mitophagy. We were surprised, however, to find that the lipid-binding domains of PRKCD were dispensable for mitochondrial recruitment. This likely suggests that PRKCD is recruited to mitochondria through protein interactions mediated by the remaining protein regions. However, it is still likely that lipid binding is important for its activity. DAG can be formed from phosphatidic acid by phosphatidic acid phosphohydrolases, such as the Lipin family of proteins. Lipin-1 deficiency is associated with the accumulation of mitochondria combined with morphological abnormalities[56]. In addition, Lipin-1 depletion was found to

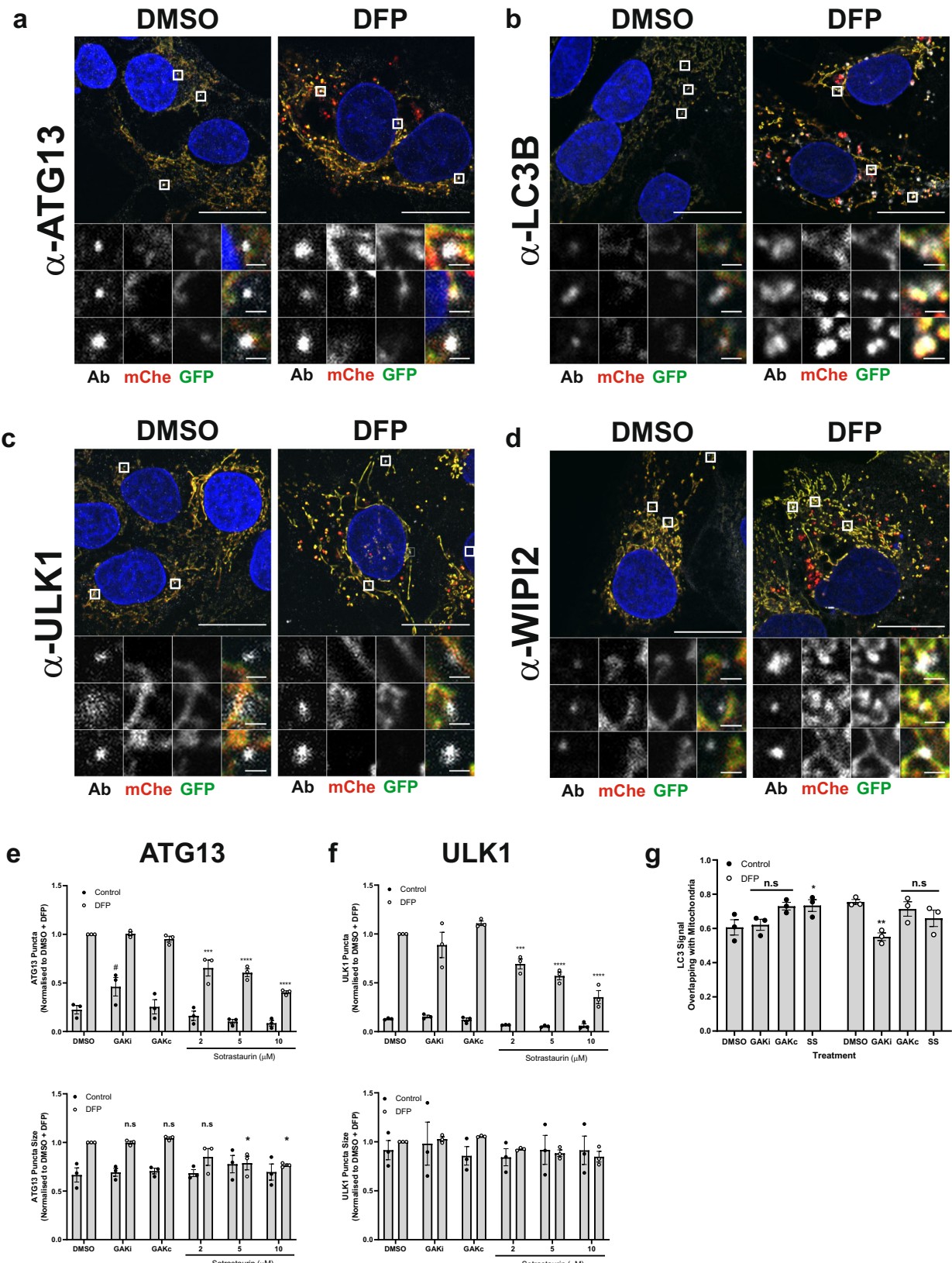

reduce the level of PKD1 phosphorylation with a subsequent decrease in VPS34 mediated PtdIns(3)P formation[56]. Further examination of PKD1 regulation may therefore be important for studying the role of PKCs in DFP-induced mitophagy. Alternately, exposure of the unique inner mitochondrial phospholipid cardiolipin to the outer membrane is a known interactor of LC3 and signal for mitophagy that is dependent upon phospholipid scramblases such as PLSCR3 for bilayer movement[57,58]. PRKCD has been found to phosphorylate and regulate PLSCR3 activity during UV irradiation and is thereby another area that warrants further investigation under DFP-inducing conditions[26].

**Fig. 6 Early autophagy protein recruitment is defective upon PKCi. a–d** U2OS IMLS cells were treated ± 1 mM DFP for 24 h, fixed and stained for nuclei (DAPI) and the indicated endogenous autophagy markers; **a** ATG13, **b** LC3B, **c** ULK1 or **d** WIPI2. Representative ×63 images of cells taken by Zeiss LSM710 are shown, scale bar = 10 µm. **e**, **f** U2OS cells were treated ± 1 mM DFP for 24 h together with GAKi (10 µM), GAKc (10 µM) or sotrastaurin (2–10 µM), then fixed in PFA before staining for nuclei (DAPI) and the indicated endogenous early autophagy markers; **e** ATG13, **f** ULK1. The number and size of puncta formed for each marker were analysed and values obtained were normalised to the DMSO + DFP control. Mean values were plotted from $n = 3$ independent experiments ± SEM. Significance was determined by two-way ANOVA and Dunnett's multiple comparisons test to the DMSO + DFP control sample. **g** U2OS cells were treated ±1 mM DFP for 24 h with GAKi (10 µM), GAKc (10 µM) or sotrastaurin (10 µM) and labelled for mitochondria with MitoTracker red. Co-localisation of LC3 signal with mitochondria was determined by Mander's co-localisation. Significance was determined by two-way ANOVA and Dunnett's post-test to the DMSO sample. Where indicated *$P < 0.05$, **$P < 0.01$, ***$P < 0.001$, ****$P < 0.0001$, n.s. = not significant and #$P < 0.05$ (to the DMSO control). For precise $P$ values, see the source data file.

Importantly, neither GAK nor PRKCD inhibition was able to modulate PRKN-mediated mitophagy. This adds further evidence that the machinery required for PRKN-dependent and independent mitophagy pathways are fundamentally different. Tantalisingly, both *C. elegans* and *D. rario* models indicate that basal mitophagy (along with DMOG-stimulated in zebrafish experiments) can be regulated by these kinases as opposed to stress-induced mitophagy, such as that regulated by PRKN. By looking at the hindbrain region of zebrafish, showing the highest expression of Prkcd, we observed a significant reduction in the level of mitophagy upon knockout of *prkcda* and *prkcdb* and impairment of locomotory responses. This may be due to deterioration of hindbrain locomotory neurons as a consequence of impaired mitophagy or may be associated with an anxiety-like response (thought to be triggered in zebrafish due to light changes) noted previously in mice to be impaired with PRKCD depletion[59].

To conclude, this initial screen of lipid-binding proteins in DFP-induced mitophagy identified two kinases that have been validated by functional characterisation and confirmation in higher organisms. This highlights the importance of protein-lipid interactions and provides a strong initial basis for further investigation into the molecular mechanisms of mitophagy.

## Methods

**Materials.** Lysotracker Red DND-99 (L7528) was from ThermoFisher Scientific. Antimycin A (A8674), DFP (379409), DMOG (D3695), SGC-GAK-1 (GAKi, SML2202), SGC-GAK-1N (GAKc, SML2203) and QVD-OPh (SML0063) were from Sigma Aldrich. Bafilomycin A1 (BML-CM110), CCCP (BML-CM124) were from Enzo Life Sciences. Enzastaurin (S1055), oligomycin A (S1478), and sotrastaurin (S2791) were from Selleckchem. MRT68921 (1190379-70-4) and VPS34-IN1 (1383716-33-3) were from Cayman Chemical. IVAP1966 (12 g) and IVAP1966 (12i) were gratefully received from the lab of Prof. Piet Herdewijn[29]. HY-19764 was gratefully received from the structural genomics consortium[30]. Bradford reagent dye (#5000006) was from Bio-Rad. 1,4-dithiothreitol (DTT, #441496P) was from VWR. Complete EDTA-free protease inhibitors (#05056489001) and phosphatase inhibitors (#04906837001) were from Roche.

**Cell lines, maintenance and induction of mitophagy.** U2OS FlpIN TRex cells (kindly received from Prof. Stephen Blacklow) with the stable dox-inducible expression of MLS-EGFP-mCherry (referred to as IMLS cells)[15] were grown and maintained in a complete medium of Dulbecco's Modified Eagle Medium (DMEM —Lonza 12–741F) supplemented with 10% v/v foetal bovine serum (FBS—Sigma Aldrich #F7524) and 100 U/ml penicillin + 100 µg/ml Streptomycin (Thermo-Fisher Scientific #15140122) in a humidified incubator at 37 °C with 5% $CO_2$. U2OS IMLS cells with stable expression of PRKN were generated by cloning of PRKN into a pLenti-III-PGK viral expression vector that was co-transfected into 293FT cells (Invitrogen) with psPAX2 and pCMV-VSVG to generate lentiviral particles, which were transduced into U2OS IMLS cells and positive cells selected with puromycin (Sigma #P7255). U2OS cells (ATCC) expressing TFEB-mCherry were generated by site-specific knock-in on the *AAVS1* locus[60]. Briefly, cells were co-transfected with plasmids encoding an *AAVS1* zinc-finger nuclease and a plasmid encoding TFEB-mCherry (9:1) with FuGENE HD Transfection reagent as per the manufacturer's instructions. Positively integrated cells were selected with puromycin and utilised for experiments.

Mitophagy was typically induced utilising 1 mM DFP by addition to cell culture media for 24 h. In the case of PRKN overexpression, CCCP was used at 20 µM for 16 h or a combination of oligomycin and antimycin A (10 and 1 µM, respectively) for 16 h. In the case of PRKN-dependent mitophagy experiments, the pan-caspase

inhibitor QVD-OPh[61] was included to reduce cell death and improve imaging quality, in accordance with previous papers studying PRKN-dependent mitophagy[62].

**Imaging and image analysis.** The initial siRNA screen, secondary siRNA screen and other experiments where indicated were carried out utilising an AxioObserver widefield microscope (Zen Blue 2.3, Zeiss) with a ×20 objective (NA 0.5). Relevant channels were imaged using a solid-state light source (Colibri 7) and a multi-bandpass filter (BP425/30, 524/50, 688/145) or individual filters. The tertiary siRNA screen was carried out utilising an ImageXpress Micro Confocal (Molecular Devices) using a ×20 objective (NA 0.45). Confocal images were taken with an LSM710 microscope or LSM 800 (Zebrafish experiments) microscope (Zen Black 2012 SP5 FP3, Zeiss) utilising a ×63 oil objective (NA 1.4) combined with a laser diode (405 nm), Ar-Laser Multiline (458/488/514 nm), DPSS (561 nm) or HeNe-laser (633 nm) for relevant fluorophore acquisition. Where noted in figure legends, an Andor Dragonfly confocal microscope (Oxford Instruments) was used with a solid-state 445/488/561/640 nm laser as appropriate for relevant fluorophore acquisition.

Identification of relevant structures by image analysis was determined using CellProfiler software (v2.8.0, The Broad Institute)[63]. In the case of IMLS cell analysis for mitophagy, red-only structures were identified by dividing the red signal by green signal per pixel following background noise reduction and weighting of the red signal to match that of the green signal in non-mitophagy inducing controls. By this method, a value of ~1 indicates "yellow" networked mitochondria and values <1 represent mitochondria that have a stronger red signal than green signal. Values of <0.5 were taken to represent true red structures, regions that are therefore twice as bright for red than green.

**Zebrafish maintenance and in situ hybridisation (ISH).** All Zebrafish experimental procedures and housing conditions followed the recommendations of the National Institute of Health Guidelines for the Care and Use of Laboratory Animals, the European Community Council Directive of November 2010 for Care and Use of Laboratory Animals (Directive 2010/63/EU) and the Norwegian Regulation on Animal Experimentation ("Forskrift om forsøk med dyr" from June 18, 2015). All experiments conducted on wild-type zebrafish and transgenic tandem-tagged mitofish larvae were done at 5dpf or earlier.

Wild-type zebrafish (AB strain) and transgenic tandem-tagged mitofish (TT-mitofish)[15] were housed at the zebrafish facility at the Centre for Molecular Medicine Norway (AVD.172) using standard practices. Embryos were incubated in egg water (0.06 g/L salt (Red Sea)) or E3 medium (5 mM NaCl, 0.17 mM KCl, 0.33 mM $CaCl_2$, 0.33 mM $MgSO_4$, equilibrated to pH 7.0). Embryos were held at 28 °C in an incubator following collection.

**Whole-mount in situ hybridisation (WISH)**
*Probe synthesis.* Whole-mount ISH for *prkcda* and *prkcdb* were performed as previously described using digoxigenin-labelled riboprobes[64]. Primer sequences for sense and antisense probes are described in Supplementary Table 2. Respective PCR products were purified and cloned into Zero Blunt TOPO PCR Cloning Kit (ThermoFisher Scientific #450245) having both SP6 and T7 promoter/primer site for in vitro RNA transcription and sequencing. Digoxigenin UTP-labelled sense and antisense RNA probe were generated by in vitro transcription of the zero blunt plasmid having the amplicon of interest, post linearisation with ECORV/BamH1 (NEB #R3195L and #R3136L), respectively, and stored at −80 °C for further use in ISH.

*In situ hybridisation (ISH) procedure.* Zebrafish larvae at 5 dpf were fixed overnight at 4 °C in 4% PFA in PBS, after which they were washed with PBST (PBS with 0.1% Tween-20) and gradually transferred to 100% methanol. Prior to WISH, the dehydrated larvae were gradually transferred back to PBST. After rehydration, the samples were treated with proteinase K to facilitate infiltration of the probes into the tissue and later washed with PBST. Next, PBST was replaced by hybridisation buffer (HYB-, 50% deionized formamide, 5× saline sodium citrate (SSC), 0.1% Tween-20, 9.2 mM citric acid) supplemented with 50 µg/ml heparin and 0.5 mg/ml

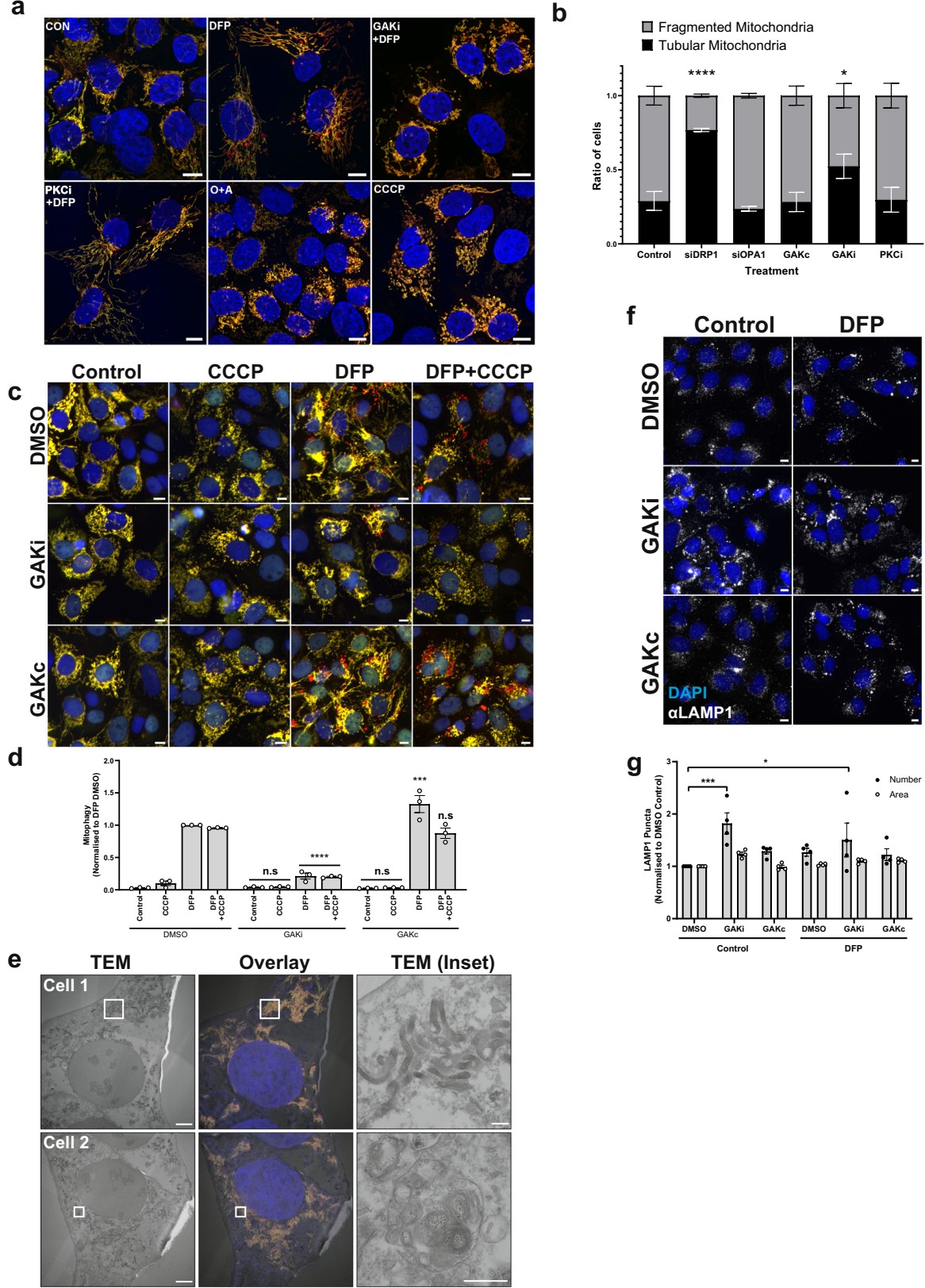

tRNA (HYB+) in a 2-ml Eppendorf tube at 70 °C for 4 h. This was followed by incubation o/n with probes diluted in HYB+ at 70 °C. After o/n probe hybridisation, samples were washed once in pre-heated HYB- for 10 min at 70 °C. Afterwards, all samples are taken through a series of pre-heated HYB- diluted in 2× SSC-T (SSC containing 0.01% Tween-20), first 75% HYB-/2×SSC-T, 50% HYB-/2×SSC-T, 25% HYB-/2×SSC-T and finally 100% 2×SSC-T at 70 °C. This was followed by two 30-min incubations in pre-heated 0.2×SSC-T at 70 °C. The samples were then subsequently taken through a graded series of 0.2×SSC-T diluted in PBST, first 75% 0.2×SSC-T/PBST, then 50% 0.2×SSC-T/PBST and finally 25% 0.2×SSC-T/PBST, all at RT. All samples were subsequently washed twice in PBST at RT for 10 min before being transferred to blocking buffer (BB, 2% sheep serum, 2 mg/ml BSA in PBST) for 4 h at RT. After blocking, all samples were incubated o/n in BB containing pre-incubated anti-digoxigenin (Roche, 1:5000) antibody while on a rocker at 4 °C.

**Fig. 7 GAKi induces abnormal mitochondrial and lysosomal morphology. a** Representative ×63 images of U2OS IMLS cells taken by Zeiss LSM710 confocal microscopy. Cells were treated ± 1 mM DFP 24 h in addition to GAKi (10 μM), sotrastaurin (PKCi—2 μM), oligomycin and antimycin A (O + A— 10 μM and 1 μM, respectively) or CCCP (20 μM), scale bar = 10 μm. **b** Machine-learning classification of U2OS IMLS cell mitochondrial network as fragmented or tubular (utilising EGFP images, see methods) ± SEM from $n = 3$ independent experiments after 24 h treatment with GAKi (10 μM), GAKc (10 μM) or sotrastaurin (PKCi—2 μM) compared to 72 h knockdown of non-targeting control, siDRP1 or siOPA1. Significance was determined by two-way ANOVA followed by Dunnett's post-test to the control treatment. **c** U2OS IMLS cells were treated as indicated with DMSO, GAKi or GAKc (10 μM each) for 24 h in addition to either DFP (24 h, 1 mM), CCCP (20 μM, 12 h) or in combination. Images obtained by ×20 objective, scale bar = 10 μm. **d** Quantitation of mean mitophagy per cell from cells treated as in (**c**) ± SEM from $n = 3$ independent experiments. Significance was determined by two-way ANOVA followed by Dunnett's post-test to the equivalent DMSO treatment. **e** U2OS IMLS cells were treated with 1 mM DFP + 10 μM GAKi for 24 h prior to fixation for CLEM analysis. EM images demonstrate mitochondrial clustering (Cell 1) and an increase in autolysosome structures (Cell 2) induced by GAKi treatment, scale bar = 10 μm, inset =1 μM. **f** U2OS cells treated ± 1 mM DFP 24 h in addition to DMSO, GAKi (10 μM) or GAKc (10 μM) were PFA fixed and subsequently stained for endogenous LAMP1. Images acquired at ×20 by widefield microscopy on a Zeiss AxioObserver microscope, scale bar = 10 μm. **g** Quantitation of LAMP1 structures identified in (**f**) for size and number and plotted as mean ± SEM from $n = 4$ independent experiments. Significance was determined by two-way ANOVA followed by Dunnett's multiple comparisons test to the DMSO control. Significance was denoted where *$P < 0.05$, ***$P < 0.01$, ****$P < 0.0001$ and n.s. = not significant in all relevant panels. For precise $P$ values, see the source data file.

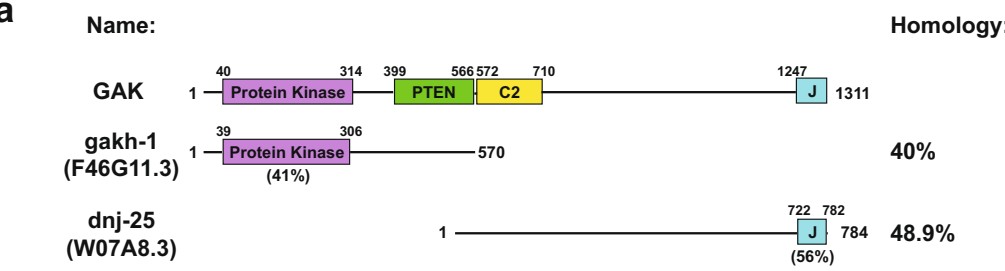

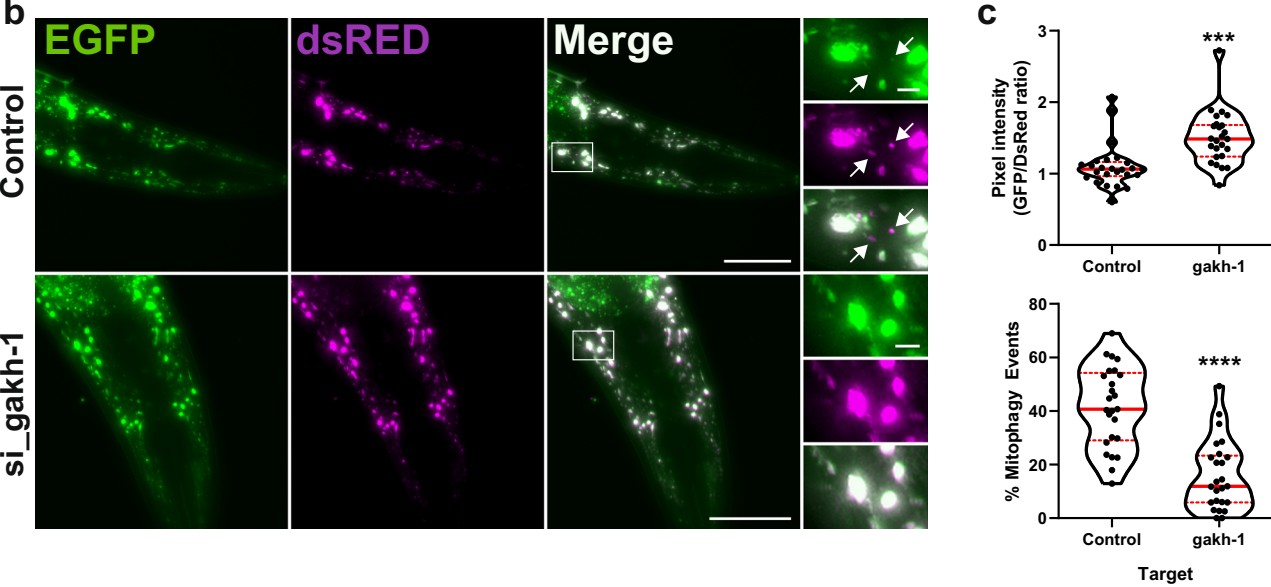

**Fig. 8 gakh-1 regulates basal mitophagy in vivo. a** Schematic representation of GAK domain structure and orthologues gakh-1 and dnj-25 present in *C. elegans*. Homology values were obtained by protein blast alignment. **b** In vivo detection of mitophagy in *C. elegans*. Transgenic nematodes expressing mtRosella in body-wall muscle cells were treated with *gakh-1* RNAi or pL4440 control vector. dsRED only structures represent mitochondria in acidic compartments (arrowheads). Scale bar = 50 μm, insets = 5 μm. Representative image from $n = 25$ worms. **c** Mitophagy stimulation signified by the ratio between pH-sensitive GFP to pH-insensitive dsRED ($n = 25$ worms per group, upper panel). Quantification of the frequency of mitochondria undergoing mitophagy (dsRED puncta lacking EGFP co-localisation) are expressed as a percentage of total mitochondria detected ($n = 25$ worms per group, lower panel). The data are presented as violin plots of individual values with median (red, solid line) and quartiles (red, dashed line) shown. Significance was determined by unpaired two-tailed $t$ test from $n = 2$ independent experiments, where **$P < 0.001$ or ****$P < 0.0001$. For precise $P$ values, see the source data file.

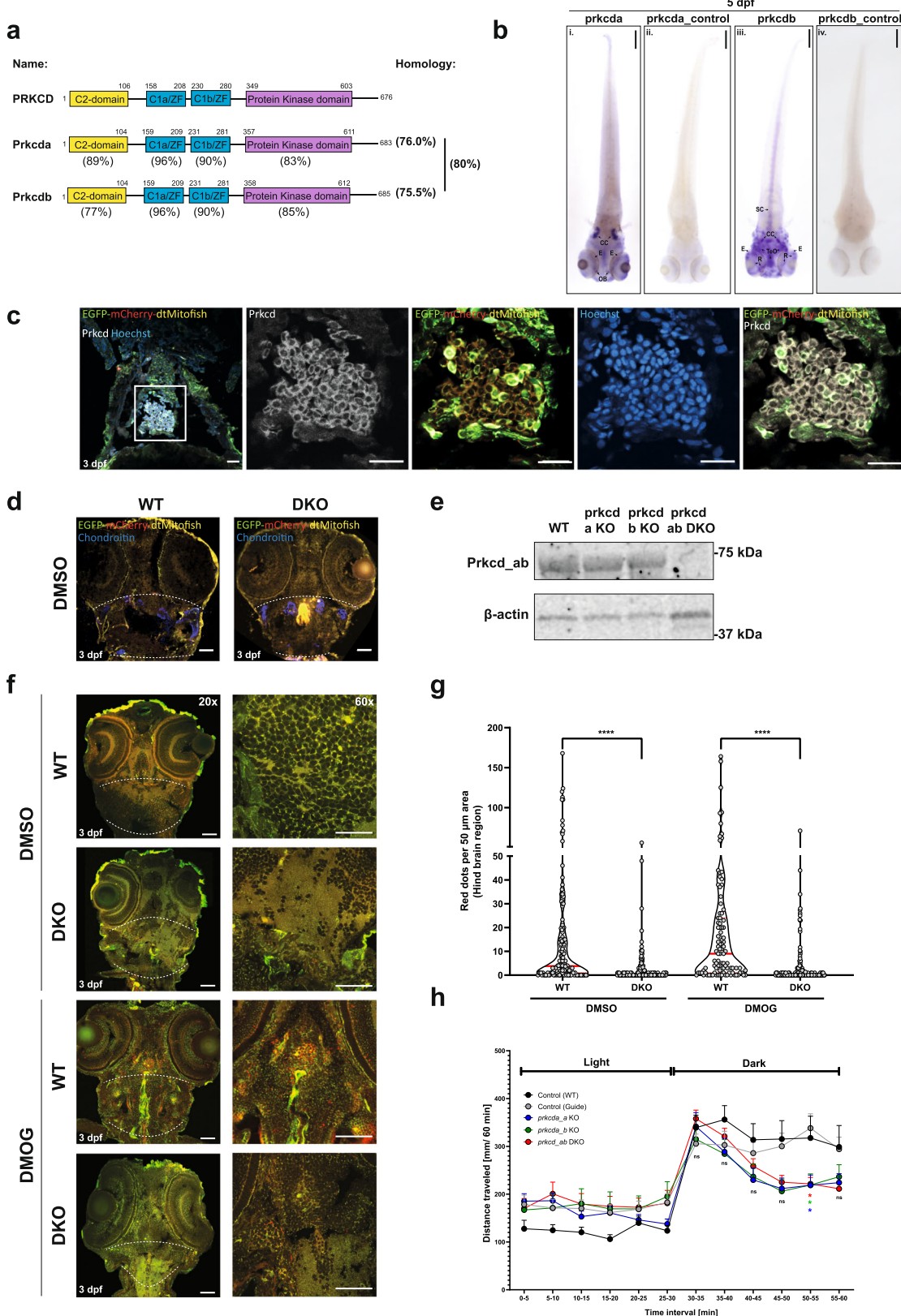

After o/n incubation, the antibody dilution was removed from all samples and replaced by PBST. Hereafter samples were washed six times with PBST for 15 min per wash while on a rocker at RT. Subsequently, all samples were washed three times in staining buffer (SB, 0.1 M Tris-HCl pH 9.5, 50 mM MgCl₂, 0.1 M NaCl, 0.01% Tween-20) at RT while on a rocker, once for 5 min and twice for 15 min. The samples were then transferred to 24-well plates and incubated with 500 µl staining buffer

supplemented with NBT (Roche, 1:200) and BCIP (Roche, 1:300) at RT. The staining reaction was stopped by adding PBST and subsequent washes with PBST were performed for 5 min at RT. Larvae were postfixed again with 4% PFA at RT for 10 min and the background was cleared using graded series of ethanol. Finally, the larval samples were preserved in 50% glycerol/50% PBS and later imaged on a Stemi 508 DOC Zeiss stereomicroscope with a mounted axiocam colour camera.

**Fig. 9 prkcda and prkcdb regulate mitophagy in vivo. a** Overview and schematic diagram of human PRKCD, zebrafish prkcda and prkcdb proteins. Percentage identity of respective domains on comparison with human counterpart shown below the zebrafish domains. Also shown is the percentage identity of the orthologues amongst each other. **b** Dorsal view of spatial expression pattern of *prkcda* and *prkcdb* at 5 dpf as demonstrated by whole-mount in situ hybridisation using a 5′UTR probe. Scale bar = 200 μm. **c** Representative confocal images of cryosections taken from transgenic tandem-tagged mitofish at 3 dpf, fixed and stained for nuclei (Hoechst) and endogenous Prkcd. The image shows the section of a larval zebrafish hindbrain. Representative ×40 image of the zebrafish hindbrain section was taken on Zeiss LSM 800, scale bar = 20 μm. **d** Representative ×20 confocal images (Zeiss LSM 800) of cryosections taken from transgenic tandem-tagged mitofish at 3 dpf, fixed and stained for chondroitin sulphate. The markings (white dashed line) represent the hindbrain region, following the chondroitin staining. Scale bar = 50 μm. **e** Representative immunoblots of Prkcd and β-actin on whole embryo lysates from wild-type and single or double prkcda/prkcdb KO (DKO) animals. **f** Representative ×20 and ×60 confocal images of cryosections taken from control (guide only) and prkcd_ab DKO transgenic tandem-tagged mitofish larvae treated with DMSO or with DMOG at 3 dpf for 24 h. Images are from within the hindbrain region of the respective larvae as marked (dashed white line). Scale bars = 50 μm. **g** Quantitation of the number of red puncta from a 50-μm hindbrain region (as marked in **f**) from control and prkcd_ab DKO tandem-tagged mitofish larvae. Plots demonstrate data distribution and median value (red line). Significance was determined by two-way ANOVA followed by Tukey's post-test to compare all groups. **h** Motility analysis of zebrafish embryos at 5 dpf using the "Zebrabox" automated videotracker (Viewpoint). The assay was carried out during daytime and consisted of one cycle of 30 min exposure to light followed by 30 min of darkness. Data represent mean distance moved ± SEM. Each group consisted of 43–124 larvae from *n* = 9 independent experiments. Significance was determined by two-way ANOVA followed by Dunnett's post-test to the control (WT). Significance are denoted where *$P < 0.05$, **$P < 0.01$, ***$P < 0.001$, ****$P < 0.0001$, n.s. = not significant. For precise *P* values, see the source data file.

**Screening library**. Human targets for the siRNA library were identified by using the ExPASY PROSITE sequence motif database identifier for human proteins containing true-positive identified lipid-binding domains. These included C1 domains (ID: PDOC00379), C2 domains (ID: PDOC00380), ENTH (ID: PDOC50942), PH Domain (ID: PDOC50003), PX Domain (ID: PDOC50195), FYVE domain (PDOC50178), GRAM or PROPPIN (SVP1 family) domains (No ID). This list was cross-checked against several previously published U2OS cell line proteomes (determined by mass spectrometry) and proteins not observed to be expressed in U2OS cells were removed[19,20]. See Supplementary Data 2 for a full list of siRNA targets.

**siRNA knockdown**. The primary screen was carried out using a pooled siRNA approach with three Silencer Select siRNA oligonucleotides targeting each gene at 2.5 nM final concentration each (7.5 nM final). For transfection, 125 μl of Opti-MEM (Thermofisher Scientific #31985070) containing 100 ng/ml doxycycline (Clontech #631311) and 0.1 μl RNAiMAX per pmol siRNA (Thermofisher Scientific #13778150) was added to each well of an Ibidi 96-well μ-plate (Ibidi #89626). After 5 min at room temperature (RT), 25 μl of 75 nM siRNA (pooled) diluted in OptiMEM was added per well and incubated a further 15 min at RT. U2OS IMLS cells were trypsinised and resuspended in complete media before centrifugation at 300×*g* for 5 min at RT. Media was removed and cells resuspended in OptiMEM to 2 × 10⁵ cells/ml. In total, 100 μl of cells were added per well, and samples were incubated for 16 h at 37 °C. The media was then removed and changed to complete media for a further 24 h before the media again was changed to complete media ± 1 mM DFP+ 100 ng/ml doxycyline and incubated for 24 h to induce mitophagy. Control wells with BafA1 treatment were dosed 2 h prior to fixation. At the end of the experiment, samples were washed once in PBS and then fixed in 3.7% PFA, 200 mM HEPES pH 7 for 15 min/37 °C. PFA was then quenched by washing twice and incubating a further 15 min in DMEM+ 10 mM HEPES pH 7. Wells were then washed twice with PBS and then incubated in PBS+ 2 μg/ml Hoechst to stain nuclei for a minimum of 1 h prior to imaging. Images were obtained on a Zeiss AxioObserver widefield microscope with a ×20 objective acquiring a minimum of 35 fields of view per treatment. Analysis of red-only punctate structures was carried out utilising CellProfiler from a minimum of 1000 cells per condition per replicate.

**Identification and plotting of protein–protein interactome (PPI) networks**. PPI represents the physical interaction among a set of proteins. PPI was obtained from Biological General Repository for Interaction Datasets (BioGRID) version BIOGRID-ORGANISM-3.5.185.mitab[65] (compiled April 25, 2020) containing non-redundant and curated interactions. The networks were visualised using Cytoscape (v3.8.0)[66], we considered only the connected component of these seed networks for statistical and functional analysis. Functional and pathway analysis of connected components of the interaction network was performed by ShinyGO (v0.66)[67]. We only considered GO terms for cellular components, molecular functions and biological process with significant *P* value and enrichment values. Graphs were plotted using R package ggplots.

**RNA isolation, cDNA synthesis and qPCR**. For quantifying knockdown in the secondary deconvolution siRNA screen, RNA was isolated and cDNA generated from transfected U2OS cells using *Power* SYBR Green Cells-to-C$_T$ kit (Thermo-Fisher Scientific #4402955) as per the manufacturer's instructions.

For other experiments, RNA was isolated from cells or zebrafish (~50 embryos per sample) using Trizol reagent (ThermoFisher Scientific #15596026). cDNA was synthesised from RNA using Superscript III reverse transcriptase (ThermoFisher Scientific #18080085) according to the manufacturer's instructions. Amplification

was performed with KAPA SYBR FAST qPCR Kit using a CFx96 real-time PCR system (Bio-Rad) using primers designed to amplify target genes as indicated in Supplementary Table 2 following normalisation of transcript levels to TATA-box-binding protein (TBP—cell samples) or β-actin (zebrafish samples) using the $2^{-\Delta\Delta Ct}$ method.

**Western blotting**. For western blotting experiments, cells were treated as indicated in figure legends prior to moving onto the ice and washing twice with cold PBS. Cells were lysed on ice in NP-40 lysis buffer [50 mM HEPES pH 7.4, 150 mM NaCl, 1 mM EDTA, 10% (v/v) glycerol, 0.5% (v/v) NP-40 + 1 mM DTT, 1× phosphatase inhibitors and 1× protease inhibitors fresh] and incubated 5 min prior to collecting. For zebrafish samples, embryos were collected at 3 dpf and lysed in RIPA buffer [50 mM Tris-HCl pH 8, 150 mM NaCl, 5 mM EDTA, 1% NP-40, 0.5% sodium deoxycholate, 0.1% SDS, 1× protease inhibitor cocktail], ~20–30 embryos were used per gel lane.

Samples were clarified by centrifugation at 21,000 × *g*/4 °C/10 min and supernatant retained. Protein levels were quantified by Bradford assay (Bio-Rad #5000006) relative to a BSA standard. Samples were normalised and added to loading sample [1× = 62.5 mM Tris pH 6.8, 10% (v/v) glycerol, 2% (w/v) SDS, 0.005% (w/v) bromophenol blue] to achieve 30–50 μg of protein per lane. Samples were run by acrylamide gel and transferred to PVDF (350 mA/50 min). Samples were blocked in TBS Odyssey Blocking Buffer (Li-Cor #927-50000) for 30 min/RT before incubation overnight at 4 °C with primary antibodies (TBS blocking buffer + 0.2% Tween). Membranes were washed 3 × 10 min in TBST before secondary antibody incubation (TBS blocking buffer + 0.2% Tween + 0.01% SDS) for 1 h. Membranes were washed 3× 10 min with TBST before a final wash in TBS only and membrane imaging.

**Antibodies**. Primary antibodies targeting ATG13 (#13468, Clone E1Y9V, IF = 1:250, WB = 1:1000), AMPK P-T172 (#2535, 1:1000), β-actin (#3700, Clone 8H10D10, 1:5000), BNIP3 (#44060, Clone D7U1T, 1:1000), BNIP3L (#12396, Clone D4R4B, 1:1000), COXIV (#4850, Clone 3E11, WB = 1:1000), p70 (#9202, WB = 1:1000), p70 T389 (#9205, WB = 1:1000), PDH (#2784, WB = 1:1000), PRKCD (#9616, D10E2, IF = 1:250, WB = 1:1000), PRKCD P-S663 (#9376, WB = 1:1000), LC3B (western blotting only, #3868, Clone D11, WB = 1:1000), ULK1 (#8054, IF = 1:250, WB = 1:1000), ULK1 P-S555 (#5869, WB = 1:1000), ULK1 P-S757 (#6888, Clone D1H4, WB = 1:1000) were from Cell Signaling Technology. Chondroitin sulphate (#Ab11570, Clone CS-56, IF = 1:1000), FUNDC1 (#Ab74834, WB = 1:1000), GAK (#Ab115179, Clone 1C2, WB = 1:500), PRKCD (#Ab182126, Clone EPR17075, IF = 1:100), NIPSNAP1 (#Ab67302, WB = 1:1000), MTCO2 (#Ab110258, Clone 12C4F12, WB = 1:1000) and WIPI2 (#Ab105459, Clone 2A2, IF = 1:500, WB = 1:2000) were from Abcam. ATG13 P-S318 (#600-401-C49S, WB = 1:1000) was from Rockland. LAMP1 (sc-20011, Clone H4A3, IF = 1:1000) and TOM20 (sc-17764, Clone F-10, WB = 1:1000) were from Santa Cruz Biotechnology. α-Tubulin (T5168, Clone B-5-1-2, WB = 1:5000) was from Sigma Aldrich. TIM23 (611223, Clone 32, IF = 1:500, WB = 1:1000) was from BD Biosciences. LC3B (Immunofluorescence only, PM036, IF = 1:500) was from MBL International.

Secondary antibodies for western blotting are indicated in the source data file, these included anti-rabbit (Starbright Blue, Bio-Rad, 12004161, 1:5000) (DyLight 800, ThermoFisher Scientific, SA5-10044) (DyLight 680, ThermoFisher Scientific, SA5-10042, 1:10000) or anti-mouse (Starbright Blue, Bio-Rad,12004158, 1:5000) (DyLight 680, ThermoFisher Scientific, SA5-10170, 1:10000) or anti-tubulin (hFAB Rhodamine, Bio-Rad, 12004166, 1:1000). Secondary antibodies for immunofluorescence were used at 1:1000 and included anti-rabbit Alexa Fluor-594

(Invitrogen, A11058), Alexa Fluor-647 (ThermoFisher Scientific, A21245) or anti-mouse Alexa Fluor-647 (ThermoFisher Scientific, A21236).

**Phos-tag gels**. Phos-tag acrylamide gels were prepared in line with the manufacturer's instructions. Briefly, 8% resolving polyacrylamide gels were prepared containing 25 μM Phos-tag reagent (Wako Chemicals #AAL-107) and 50 μM MnCl$_2$[34]. Samples to be run for analysis were diluted in a loading sample containing 10 mM MnCl$_2$. Acrylamide gels were ran at 40 mA until complete and washed 3 × 10 min/RT in 1× transfer buffer (48 mM Tris, 39 mM glycine, 0.0375% (w/v) SDS) + 10 mM EDTA followed by 1 × 10 min in 1× transfer buffer. Samples were then transferred to PVDF at 350 mA/50 min and treated as noted earlier for western blot samples.

**PFA fixation, antibody staining and imaging**. Cells to be imaged were seeded onto glass coverslips 16 h prior to treatments as indicated in figure legends. Following treatment, cells were washed once with PBS prior to addition of warmed fixation buffer (3.7% (w/v) PFA, 200 mM HEPES pH 7.4) or for double-tag IMLS cells (3.7% (w/v) PFA, 200 mM HEPES pH 7) and incubated 15 min at 37 °C. Coverslips were washed twice and incubated 1 × 15 min with DMEM + 10 mM HEPES pH 7.4 (IMLS = pH 7). Cells were then washed once with PBS and then permeabilised by incubation for 5 min with permeabilisation buffer (0.2% (v/v) NP-40 in PBS). Cells were washed twice and then incubated 20 min with IF blocking buffer (PBS + 1% (w/v) BSA) to block the samples. Coverslips were then incubated 1 h/37 °C with primary antibodies diluted in IF blocking buffer before washing 3 × 10 min in IF blocking buffer. Coverslips were then incubated 30 min/RT with appropriate secondary antibodies. Finally, samples were washed 3 × 10 min in IF blocking buffer prior to mounting onto coverslides with ProLong Diamond Antifade Mountant with DAPI (ThermoFisher Scientific #P36962). Slides were allowed to cure overnight before imaging with either a Zeiss AxioObserver widefield microscope (×20) or Zeiss LSM 800 confocal microscope (×60).

**Citrate synthase assay**. To biochemically quantify mitochondrial abundance, we assayed citrate synthase activity from cell lysates. Briefly, U2OS cells were grown and subject to treatments as described in figure legends, cells were then washed twice with PBS on ice before lysis [50 mM HEPES pH 7.4, 150 mM NaCl, 1 mM EDTA, 10% glycerol, 0.5% NP-40, 1 mM DTT, 1× phosphatase inhibitors, 1× protease inhibitors]. Cell lysates were clarified by centrifugation at 21,000 × g/ 10 min/4 °C and supernatants retained. Protein concentration was determined by the Bradford assay. To determine citrate synthase activity, 1 μl of protein lysate was added to 197 μl of CS assay buffer [100 mM Tris pH 8, 0.1% Triton X-100, 0.1 mM acetyl CoA, 0.2 mM DTNB [5,5'Dithiobis(2-nitrobenzoic acid)]] in a multiwell plate. At the assay start point, 2 μl of 20 mM Iodoacetamide was added per well and incubated at 32 °C and reactions monitored at $\lambda_{Abs} = 420$ nm for 30 min in a FLUOstar OPTIMA (v2.20R2, BMG Labtech) plate reader and compared to iodoacetamide null controls. The $\Delta \lambda_{Abs}$ was plotted and the reaction rate determined across the linear range before saturation. The reaction rate was then normalised to the protein concentration and plotted relative to the control.

**Correlative light electron microscopy (CLEM)**. For CLEM, U2OS IMLS cells were grown on photo-etched coverslips (Electron Microscopy Sciences, Hatfield, USA). The next day, cells were treated with DFP (1 mM) ± GAKi (10 μM) for 24 h. Cells were then fixed in 4% formaldehyde, 0.1% glutaraldehyde/0.1 M PHEM (60 mM PIPES, 25 mM HEPES, 2 mM MgCl$_2$, 10 mM EGTA, pH 6.9) for 1 h. The cells were mounted with Mowiol containing 2 μg/ml Hoechst 33342 (Sigma Aldrich). Mounted coverslips were examined with a Zeiss LSM710 confocal microscope with a Zeiss plan-Apochromat ×63/1.4 Oil DIC III objective. Cells of interest were identified by fluorescence microscopy and a Z-stack was acquired. The relative positioning of the cells on the photo-etched coverslips was determined by taking a DIC image. The coverslips were removed from the object glass, washed with 0.1 M PHEM buffer and fixed in 2% glutaraldehyde/0.1 M PHEM for 1 h. Cells were postfixed in osmium tetroxide and uranyl acetate, stained with tannic acid, dehydrated stepwise to 100% ethanol and flat-embedded in Epon. Serial sections (~100–200 nm) were cut on an Ultracut UCT ultramicrotome (Leica, Germany), collected on formvar coated slot-grids. Samples were observed in a Thermo Scientific™ Talos™ F200C microscope and images were recorded with a Ceta 16 M camera. For tomograms, single-tilt image series were recorded between −60° and 60° tilt angle with 2° increment. Single-axis tomograms were computed using weighted back projection and, using the IMOD software package version 4.9[68].

**Mitochondrial enrichment**. Cells to be enriched for mitochondria were grown and treated as noted in figure legends. Cells were then moved to ice and washed twice with ice-cold PBS. In all, 1 ml of mito fractionation buffer (5 mM Tris-HCl pH 7.5, 210 mM mannitol, 70 mM sucrose, 1 mM EDTA pH 7.5, 1 mM DTT, 1× protease and phosphatase inhibitors) was added per 10-cm dish and scraped to collect cells. A "cell homogeniser" (Isobiotec) was utilised with a 16-μm clearance ball and prepared by passing through 1 ml of mito fractionation buffer. The cell solution was collected in a 1 ml syringe and passed through the cell homogeniser nine times.

The resulting mix was centrifuged 500 × g/4 °C/5 min to pellet unbroken cells and nuclei. The supernatant was taken, and a small sample retained as post-nuclear supernatant, the remaining was centrifuged at 10,000 × g/4 °C/10 min to pellet mitochondria. The supernatant was removed to waste, and the pellet resuspended in 500 μl mito fractionation buffer and 10,000 × g/4 °C/10 min centrifugation step repeated. The supernatant was removed once more, the pellet represents enriched mitochondria that could be added directly to the protein loading sample for downstream western blotting.

**Mitochondrial classifier**. To classify mitochondrial phenotypes, mitochondrial features were first identified in CellProfiler (v2.8.0, The Broad Institute) from EGFP images using the U2OS IMLS cell line. Subsequently, CellProfiler Analyst (v2.2.1, The Broad Institute) was used to generate two classifications bins based upon siDRP1 or siOPA1 treated samples as training sets (to represent fragmented or hyperfused phenotypes respectively). Key features were identified using a random forest model and led to a confusion matrix of >0.90 for each phenotype. These classification parameters were then applied to all samples and cells scored for hyperfused or fragmented phenotype and plotted as a percentage of the overall population.

**Crystal violet staining**. U2OS cells were seeded into 96-well plates in triplicate at 2 × 10$^4$ cells per well and incubated overnight in complete media. Cells were then treated for 24 h with indicated compounds and doses, utilising puromycin as a positive control. Following treatment, cell media was removed and cells were washed twice with a gentle stream of water. This was then removed and 100 μl of staining solution (0.5% (w/v) crystal violet, 20% methanol) was added and incubated 20 min/RT with gentle rocking. Wells were washed 4× with water, all liquid removed and left overnight to air dry. Then 200 μl per well of 100% methanol for 20 min/RT was added with gentle rocking and sample absorbance read at OD$_{570}$. Sample values were adjusted by no-well control (blank) wells and viability was determined by normalisation to untreated control.

**Quantification of mitophagy in C. elegans**. The strain used to monitor mitophagy process in C. elegans was IR2539: unc-119(ed3);Ex[$_{pmyo}$.3TOMM-20::Rosella;unc-119(+)]. Standard procedures for C. elegans strain maintenance were followed. Nematode rearing temperature was kept at 20 °C. For RNAi experiments, worms were placed on NGM plates containing 2 mM IPTG and seeded with HT115(DE3) bacteria transformed with either the pL4440 vector or the gakh-1 RNA construct for two generations. Synchronous animal populations were generated by hypochlorite treatment of pregnant adults to obtain tightly synchronised embryos that were allowed to develop into adulthood under appropriate, defined conditions. Progeny of these adults were tested on adult day 2. We performed imaging of mitophagy process in C. elegans based on the methods we had established[69–71]. Briefly, worms were immobilised with levamisole before mounting on 2% agarose pads for microscopic examination using EVOS Imaging System. Images were acquired as Z-stacks under the same exposure. Intensity thresholds were set to detect mitochondrial structures without size restrictions. Average pixel intensity values and the number of GFP/DsRed puncta were calculated by image analysis of different animals using FIJI.

**CRISPR/Cas9 genome editing in zebrafish and microinjections**. To generate prkcda and prkcdb knockout embryos, we utilised CRISPR/Cas9 as described earlier[72]. Briefly, potential gRNA target sites were identified using the online web tool CRISPR Design (http://CRISPR.mit.edu) or CHOPCHOP (http://chopchop.cbu.uib.no/index.php)[73]. Genomic DNA sequences retrieved from Ensembl GRCz10 and z11 (http://uswest.ensembl.org/Danio_rerio/Info/Index) were used for the target site searches. Three guide RNAs were designed each for prkcda and prkcdb, respectively, based on predictions from the aforementioned web programs. All sgRNAs were prepared by in vitro transcription of double-stranded deoxyoligonucleotide templates. The target-specific sgRNA primer is annealed to the universal primer using phusion polymerase (ThermoFisher Scientific, #F530L) in a process called fill-in PCR. The resultant double-stranded deoxyoligonucleotide product (131 bp) is gel purified and becomes the template for in vitro transcription reaction performed by using AmpliScribe-T7 flash transcription kit (Lucigen, #ASF3257). The resultant oligo is the sgRNA which is precipitated in 100% ethanol and stored at -80 until further use. Cas9 nuclease (EnGen Cas9 NLS, NEB) was combined with an equimolar mixture of 3× sgRNA's (or 6× for prkcd_ab DKO) and incubated for 5–6 min at room temperature. After incubation, the mixture was immediately placed back on the ice, until pipetted into the capillary needle used for microinjection and then approximately 1 nl of 5 μM sgRNA:Cas9 complex was microinjected into the cytoplasm of the one-cell-stage zebrafish embryo.

Oligonucleotides used for sgRNA synthesis are listed in Supplementary Table 2, a universal primer was used with individual sgRNA primers (5'-AAAAGCACCGACTCGGTGCCACTTTTTCAAGTTGATAACGGAC-TAGCCTTA TTTTAACTTGCTATTTCTAGCTCTAAAAC-'3).

**Zebrafish locomotor assay**. Larval motility was monitored using the ZebraBox and Viewpoint software (v3.10.0.42, Viewpoint Life Sciences Inc) under infrared light. At 5 days post fertilisation (dpf), larvae were singly placed in 48-well plates

with 300 μl of fish water per well, followed by incubation at 28.5 °C on a normal light cycle overnight. All experiments were completed in a quiet room at 5 dpf between 10 AM and 2 PM. Larvae were allowed to acclimate in the ZebraBox measurement apparatus for 2 h before recording. Larvae were then exposed to alternating cycles of infrared light and dark, every 30 min as described[74]. Larval locomotion was tracked with the Viewpoint software. Motility was defined as tracks moving less than 10 cm/s, but more than 0.1 cm/s.

**Zebrafish cryosectioning, immunohistochemistry, confocal microscopy and image analysis**. Zebrafish mitophagy experiments were conducted on tt-mitofish (control) and on tt-mitofish with relevant *prkcd_a/b* KO lines as described above. To examine mitophagy, zebrafish larvae were treated with 100 μM DMOG or control for 24 h. At experimental endpoints, larvae were washed once in embryo water and fixed with 4% PFA (in HEPES, pH 7–7.2) at 4 °C overnight. Post fixation, larvae were washed three times in PBS. The larvae were then cryopreserved in a 2-mL tube in increasing amounts of sucrose in 0.1 M PBS with 0.01% sodium azide. Cryopreservation was done first in 15% sucrose solution for 1 h at RT or up until larvae dropped to the bottom of the tube and then in 30% sucrose solution at 4 °C overnight with gentle shaking. Cryopreserved larvae were oriented in a cryomold (Tissue-Tek Cryomold, Sakura, Ref: 4565) with optimal cutting temperature compound (OCT compound, Tissue-Tek Sakura, Ref: 4583). Larvae were oriented with the ventral side down, and additional OCT was added to fill the mold and frozen on dry ice. A solid block of OCT with larvae oriented in the desired way was taken out from the mold and 12-μm coronal slices were sectioned on a cryostat (Thermo Scientific). Sections were collected on Superfrost Plus glass slides (Thermo Scientific, Ref: J1800AMNZ) and kept at RT for at least 2 h to firmly tether slices onto the glass slide.

The pH of all solutions and buffers used were 7–7.2. Slides were rehydrated three times in PBS at room temperature for 3 min each. The area of interest was circled by a hydrophobic PAP pen (Abcam, ab2601), and the slides were placed in a humidified chamber to avoid drying out. The sections were subsequently permeabilised with 1% Triton X-100 for 30 min and then blocked with BGT buffer (3% BSA, 0.1% Triton, 1× PBS) for 1 h at room temperature. Sections were then incubated with primary antibody solution (chondroitin—1:700; PRKCD—1:300 in 5% BSA/PBST solution) overnight at 4 °C. Post overnight incubation, slides were washed three times in PBS for 5 min each in a coplin jar and then incubated with secondary antibody solution (Alexa Fluor 405 goat anti-mouse for chondroitin and Alexa Fluor-647 goat anti-rabbit for Prkcd, dilution 1:300) along with 1 μg/ml Hoechst solution (both in PBS) for 1 h at room temperature. Finally, slides were washed three times in PBS for 5 min each and mounted using ProLong Diamond Antifade Mountant (Invitrogen, P3696). Coverslips were carefully placed over the sections. Confocal images were obtained using an Apochromat ×20/0.8, ×40/1.0 or ×63/1.2 oil DIC objective on an LSM 800 microscope (Zeiss). Image analysis was performed using CellProfiler. Segmentation of mitochondrial network was performed and mitochondrial structures were filtered as "yellow" or "red-only" based on the ratio between their EGFP and mCherry integrated intensities. The mean number of red-only structures from four grids of 250 × 250 (50 μm) pixels was used as a mitophagy readout.

**Mass spectrometry**. Cells were lysed in RIPA buffer containing Halt[TM] protease and phosphatase inhibitor cocktail (ThermoFisher Scientific, #78446) and further homogenised with a sonicator (30 s × three times with 30 s of the interval), insoluble material was removed by centrifugation. Protein concentrations were determined by BCA assay (Pierce). For each replicate, 600 μg or 30 μg of protein was used for phosphoproteomic analysis or proteomic analysis, respectively. Samples were reduced, alkylated and further digested with trypsin by FASP (filter-aided sample preparation) method. Digested peptides were transferred to a new tube, acidified and the peptides were de-salted using Oasis cartridges for STY peptides enrichments. Phosphorylated peptides enrichment was performed based on TiO$_2$[75]. Enriched peptides fractions were de-salted by C$_{18}$-stage tips.

LC-MS/MS: Dried peptide samples (from 600 μg or 30 μg of starting material for phospho-peptide or proteome respectively) were dissolved in 10 μl 0.1% formic buffer and 30% of the final yield was loaded for each MS analysis. The Ultimate 3000 nano-UHPLC system (Dionex, Sunnyvale, CA, USA) connected to a Q Exactive mass spectrometer (ThermoElectron, Bremen, Germany) equipped with a nanoelectrospray ion source was used for analysis. For liquid chromatography separation, an Acclaim PepMap 100 column (C18, 3-μm beads, 100 Å, 75-μm inner diameter) (Dionex, Sunnyvale, CA, USA) capillary of 50-cm bed length was used. A flow rate of 300 nL/min was employed with a solvent gradient of 3–35% B in 220 min, to 50% B in 20 min and then to 80% B in 2 min. Solvent A was 0.1% formic acid and solvent B was 0.1% formic acid/90% acetonitrile.

The mass spectrometer was operated in the data-dependent mode to automatically switch between MS and MS/MS acquisition. Survey full-scan MS spectra (from m/z 400–2000) were acquired with the resolution R = 70,000 at m/z 200, after accumulation to a target of 1e6. The maximum allowed ion accumulation times were 100 ms. The method used allowed sequential isolation of up to the ten most intense ions, depending on signal intensity (intensity threshold 1.7e4), for fragmentation using higher collision-induced dissociation (HCD) at a target value of 10,000 charges and a resolution R = 17,500. Target ions already selected for MS/MS were dynamically excluded for 30 s. The isolation window was m/z = 2 without

offset. The maximum allowed ion accumulation for the MS/MS spectrum was 60 ms. For accurate mass measurements, the lock mass option was enabled in MS mode and the polydimethylcyclosiloxane ions generated in the electrospray process from ambient air were used for internal recalibration during the analysis.

**Data analysis**. Raw files from the LC-MS/MS analyses were submitted to Max-Quant (v1.6.1.0) software for peptide/protein identification[76]. Parameters were set as follow: Carbamidomethyl (C) was set as a fixed modification; protein N-acetylation and methionine oxidation as variable modifications and PTY. A first search error window of 20 ppm and mains search error of 6 ppm was used. Minimal unique peptides were set to one, and FDR allowed was 0.01 (1%) for peptide and protein identification. The Uniprot human database (September_2018) was used. Generation of reversed sequences was selected to assign FDR rates. MaxQuant output files (proteinGroups.txt for proteomic data and STY(sites).txt for phosphoproteomic data) were loaded into the Perseus software[77]. Identifications from potential contaminants and reversed sequences were removed and intensities were transformed to log2. Identified phosphorylation sites were filtered only for those that were confidently localised (class I, localisation probability ≥0.75). Next, proteins identified in two out of three replicates were considered for further analysis. All zero intensity values were replaced using noise values of the normal distribution of each sample. Protein or STY abundances were compared using LFQ intensity values and a two-sample Student's T test (permutation-based FDR correction (250 randomisations), FDR cut-off: 0.05, S0: 0.1).

The complete datasets have been uploaded to ProteomXchange.

**Statistics and reproducibility**. Experimental values were used for statistical analysis using Prism (v9.0.0) where indicated using analyses and post-hoc tests as indicated in figure legends. All data values come from distinct samples. Where shown, ****$P > 0.0001$, ***$P > 0.001$, **$P > 0.01$, *$P > 0.05$ or n.s. = not significant.

In the case of representative immunoblots or fluorescence imaging, all data come from at least $n = 3$ independent experiments unless noted here. CLEM experiments (Fig. 1c) was representative from $n = 4$ cells, with several red structures correlated per cell. CLEM data in Fig. 7e is representative of structures observed in $n = 4$ cells. Mitochondrial fractionations with PRKCD lipid-binding mutant (Fig. 3h) was from $n = 2$ experiments. Cells in Fig. 6a–d are representative from >5 fields of view per antibody combination. *C. elegans* imaging (Fig. 8b) was representative from $n = 25$ worms. Zebrafish antibody staining experiments (Fig. 9c, d) were from 4+ larvae that were sectioned into 15 + 14-μm sections before antibody staining. Zebrafish mitophagy data (Fig. 9f) were representative from $n = 146–179$ random hindbrain regions. Data in Supplementary Figs. S3e, S6f and S7b are representative from $n = 2$ independent experiments.

**Reporting summary**. Further information on research design is available in the Nature Research Reporting Summary linked to this article.

## Data availability

Data supporting the findings of this study are available within the paper and supplementary information files, and from the corresponding authors upon reasonable request. The proteomics data generated in this study have been deposited in the PRIDE database under accession code [PXD022773]. ExPASY PROSITE was used to identify proteins containing specific domain sequences using the following identifiers: C1 domains: PDOC00379, C2 domains: PDOC00380, ENTH: PDOC50942, PH Domain: PDOC50003, PX Domain: PDOC50195, FYVE domain: PDOC50178. Source data are provided with this paper.

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

## Acknowledgements

We would like to thank Coen Campsteijn for assistance with live-cell imaging. We would also like to thank the Simonsen lab for their support and critical discussion throughout. We acknowledge the Norwegian Core Facility for Human Pluripotent Stem Cells at the Norwegian Center for Stem Cell Research for the use of their mycoplasma testing service, the Proteomics Core Facility at Oslo University Hospital for mass spectrometry analysis, and the Advanced Electron Microscopy Core Facility at the University of Oslo. This work was supported by the Norwegian Cancer Society (Project: 171318) and the Research Council of Norway through its Centres of Excellence funding scheme (Project: 262652) and FRIPRO grant (Project: 221831).

## Author contributions

Experimental planning, data analysis and writing of the manuscript were performed by M.J.M. and A.S. with input from all authors. B.J.M. carried out *D. rario* experiments, Seb.S carried out CLEM experiments. Y.A. and E.F. carried out *C. elegans* experiments. Sac.S. and J.W. prepared samples for MS analysis. Sak.S. carried out the analysis of interaction networks. A.H.L. generated the IMLS cell line and constructs. M.J.M., L.T.M., M.Y.W.N. and L.R.dlB. performed all remaining experiments.

## Competing interests

M.J.M. is now an employee of AstraZeneca plc. The remaining authors declare no competing interests.
