## [Peer Review File · Nature Communications]

GAK and PRKCD are positive regulators of PRKN-independent mitophagyREVIEWER COMMENTS

Reviewer #1 (Remarks to the Author):

Title: GAK and PRKCD are positive regulators of PRKN-independent mitophagy

Authors: M Munson et al

Comments:

The manuscript by Munson and co-authors represents an effort to describe the role of cyclin G-associated kinase GAK and protein kinase C delta PRKCD in PRKN-independent mitophagy. The authors showed the importance of the above mentioned kinases both in vitro and in vivo, in their manuscript. The manuscript shows interesting results, however my comments will be related to my domain of expertise.

Question: Mass Spectrometry

It is unclear to me how much total proteins were used in the mass spectrometry experiment and how was the experiment designed. Authors say here: 'For each replicate (#?, if this is 2 like the Figure 8 suggests, this is not enough, at least 3-4 biological replicates are needed, in order to talk about the abundance changes, and p-values), 600 µg of protein samples for phosphoproteomics and 30 µg for whole cell lysate proteomics were reduced..' Authors should modify this part, and better explain their idea, as there is no such thing as 'whole cell lysate proteomics'. Also, when authors say 'Cells were dissolved' this is incorrect, as cell cannot be dissolved in RIPA buffer, but the proteins can be lysed. So the authors could say: After cell lysis with RIPA buffer, proteins were measured with BCA kit. For the proteomic analysis of the whole cell lysate, this much proteins were used. For the proteomic analysis of the phosphoproteins from the cell lysate, this much proteins were used. Otherwise, it sounds confusing.

Another important thing is whether authors used any kind of protease and phosphatase inhibitors in their RIPA buffer.

Also, authors should say how much peptides they loaded for LC-MS/MS, rather than saying 3 µl.

Authors should provide details about the database used: how many entries, and what was the date the database was downloaded from Uniprot.

Reviewer #2 (Remarks to the Author):

There is a major gap in our knowledge of the factors which drive basal mitophagy and hypoxia induced mitophagy. In this study, the authors use an image-based RNAi screen to identify lipid binding proteins which play a role in mitophagy, leading to the identification of the lipid binding kinases GAK and PRKCD. The authors show that GAK and PRKCD are specific for hypoxia induced mitophagy but not Parkin mediated mitophagy nor starvation induced autophagy. This was an interesting distinction which highlights the specificity of these kinases for a specific mitophagy pathway. Using animal models, the authors show that GAK and PRKCD play a role in basal mitophagy in vivo. In addition, while a partial mechanistic insight for a role for PRKCD in mitophagy was shown, the role played by GAK remains largely unclear. I don't think this is a major weakness for the study, since detailed mechanistic insights will likely be the focus of future studies, I do think it is important to provide a bit more of an understanding of how Gak and PRKCD drive mitophagy. For example,, given the focus on lipid binding proteins in the screen, it would have been beneficial to determine whether lipid binding is necessary for the mitophagic roles of GAK and PRKCD. Additional suggestions to help partially address mechanism and strengthen the author's conclusions are provided in the comments below. Overall, this is an excellent study which makes a substantial and important advance to the fields of mitophagy and mitochondrial biology. The data are predominantly clear and convincingly support the author's conclusions. It is also clear that the discoveries made by Munson et al will be of interest to a broad readership and therefore highly suitable to the journal.

Specific comments:

1. Figure 3E: Can GAK colocalization with mitochondria during mitophagy be addressed via ectopic expression of the protein? Although the western blotting does not show evidence of mitochondrial localisation, it would be more convincing to address it via imaging since not all proteins which translocate to mitochondria are stably associated following isolation.

2. Figure 4E: The western blot data are not entirely convincing given that even in controls there is little mitochondrial protein degradation. Can the authors quantify the results or alternatively try a longer DFP treatment (up to 48h)? There are typically much higher levels of mitophagy at 48h than at 24h DFP treatment and therefore the longer treatment will help to provide clearer results. Alternatively, the quantitative MS data in Figure 4F were very convincing, can this analysis also be conducted for PRKCD inhibition? This is particularly important for PRKCD since the western blot data in Figure 4E are not clear and show evidence of mitophagy occurring (MTCO2 degradation) even at high concentrations of sotrastaurin.

3. Figure 6: It is notable that that LC3 is still lipidated (Figure 4E), especially with high concentrations of sotrastaurin treatment. Also, Figure 5G and 5E shows increased LC3 foci, indicating that sotrastaurin is affecting autophagy. To eliminate any off-target effects of the drug, and also to strengthen the authors conclusions, it would be beneficial to analyse Atg13 recruitment in GAK and PRKCD knockdown cells followed by rescue with an RNAi resistant construct. In addition, given that LC3B staining was shown in Figure 6B, can this analysis be extended to GAK and PRKCD inhibition? This analysis would help to address the idea put forward by the authors that GAK inhibition may reduce mitochondrial loading into autophagosomes (and help to justify the correlative mitochondrial morphology data). If this is not the case, then it would point toward an autophagosome-lysosome defect (which may correlate with the altered lysosome morphology data). It also might be worth the authors considering whether TFEB is correctly induced following GAK inhibition.

4. Given the major focus on screening for lipid binding proteins, it would be important to address whether lipid binding by GAK and PRKCD is required for mitophagy. This could be achieved by expressing lipid binding mutants (or lipid domain deletion) in RNAi cells followed by mitophagy assessment.

5. Sup Fig 7: The GAKi image has barely visible mitochondria as compared to the other samples. Can the authors clarify how mitochondrial network classification was conducted on those kinds of images?

Minor comments:

1. Line 47: The idea that Parkin mitophagy is not relevant in vivo due to it not being involved in basal mitophagy is not entirely accurate since it is largely a stress response pathway. Given this, it is unclear why there would be an expectation that Parkin would drive mitophagy under basal, non-stressed conditions. There are numerous studies showing activation on Parkin mitophagy during stress in animal models e.g. alcohol induced liver disease (PMID: 26159696), renal ischemia-reperfusion injury (PMID: 29172924), exercise (PMID: 29812989), myocardial infarction (PMID: 23152496). It would be worthwhile clarifying this statement.

2. Figure 1: The authors may wish to consider moving the data in Figure 1 to the supplement since it doesn't add to the study's main conclusions and mainly confirms the well-established paradigm of DFP induced mitophagy and the tandem fluorescent reporter as a measure of mitophagy. In addition, the data in Figure 8A, while certainly useful as a resource, were not used directly to follow up on and address the GAK inhibition mitophagy defect and would therefore be best placed in the supplement.

3. The authors may wish to avoid repetitively using descriptions such as 'interestingly' and 'interesting' and 'disappointing', sometimes it is best to leave it to the readers to decide.

Reviewer #3 (Remarks to the Author):

The manuscript "GAK and PRKCD are positive regulators of PRKN-independent mitophagy" by Munson et al. describes the results from a siRNA screen targeting lipid domain containing proteins with regards to a possible function in mitophagy. The paper describes the initial results from the screen, validation of hits – GAK and PRKCD – by small molecule inhibitors and an assessment of their potential function in mitophagy in vivo, using *C. elegans* and zebrafish as model systems. Overall, this is a very interesting paper that describes novel insights into parkin-independent mitophagy and the function of GAK and PRKCD at the molecular level. The paper is very well written and the experimental data is of high standard. I have the following concerns that should be addressed.

1. I wonder if the screen of 197 genes was carried out in the presence and also in the absence of DFP. The comment "data not shown" in line 116 seems to suggest so. It would be important to also show the screen results without DFP treatment as they might hint at basal mitophagy regulators.
2. In the analysis of mitophagy or red only puncta, is there any consideration of how cell numbers might affect the readout?
3. In supplementary figure 1, there is a high variability between the secondary and tertiary screen. While I appreciate that this can be the case for siRNA, it is still surprising that as little as twice the siRNA concentration can result in such great changes in variability. Is there any possible explanation for this? Does this also mean that potentially many hits were missed from the primary screen?
4. In Figure 3a: are significances missing for si ULK1? It seems highly significant, but there is no statistical annotation.
5. What is the dotted line in Figure 3b? Not explained in the legend.
6. The Kd values in supplementary figure 4a: were these measured in this study or is this referring to another publication?
7. It would be useful to see a quantification of the blot in Figure 4e, especially for the three genes that changed: MTCO2, COXIV, TIM23
8. Supplementary figure 5b is missing a control that shows a shift by Phos-tag.
9. Supplementary figure 6e is missing a blot for total ULK1; and is missing a control that changes the phosphorylation of ULK1.
10. There is quite a bit of follow-up work with a rather non-selective PKC inhibitor. Some of this could be further supported by siRNA knockdown of PRKCD. Otherwise, one might want to tone down any conclusions about specific effects by PRKCD. This is well done in the discussion section, but could be clearer in the results.
11. Quantification of Figure 7c is needed to interpret the data.
12. It seems to me, the quantification of 7e in 7f is not matching. There is an increase in control-GAKi compared to DMSO, which is not shown in the quantification.
13. Not sure what Figure 7g shows – it is missing an untreated control and quantification. Might be possible to delete?
14. It is not clear how Figure 8c was calculated. Does n=25 means mitochondria or cells? There are some very large blobs and some small puncta – was any threshold applied? Which ones were counted – large blobs and also puncta or just puncta?

Rebuttal NCOMMS-20-43623

"GAK and PRKCD are positive regulators of PRKN-independent mitophagy"

Editor:

Your revision should address all the points raised by our reviewers (see their reports below). In particular, we ask that you perform siRNA knockdown and rescue experiments to strengthen your conclusions (Reviewers #2 and #3) and perform LC3B staining in GAK and PRKCD-deficient cells (Reviewer #2). All other points raised by the referees regarding the strength of the data (controls, validation experiments, quantifications etc.) must also be addressed, with additional experimentation as appropriate.

We thank the editor for the opportunity and time to revise this manuscript for consideration in Nature Communications. Like many labs, we have unfortunately encountered significant delays and limitations with reagents/consumables due to the global pandemic.

We have addressed all comments and concerns raised by the reviewers in the point-by-point response below. Unfortunately, whilst we have tried very hard, it has not been possible to obtain results from rescue experiments (due to cell death and/or insufficient knockdown, as explained in detail below). We have however further strengthened our data by demonstrating that cells with stable overexpression of PRKCD demonstrate increased DFP-induced mitophagy (New Fig. S3c,d).

Beyond the experimental requests from reviewers, we have also included further data from our transgenic zebrafish mitophagy reporter model, demonstrating that Prkcd protein is present in the hindbrain region (New Fig. 9c) along with additional staining with Chondroitin as a positive marker for the hindbrain region (New Fig. 9d). Furthermore, we have included additional experimental data to strengthen the claims of the *in vivo* mitophagy phenotype observed (Fig. 9e,f). We have also updated Fig. 3e to include better resolution and higher quality images of PRKCD co-localisation with mitochondria in cell models.

To assist understanding, we have summarised the changes we have made to our manuscript figures since the previously submitted version:

Old Version	New Version	Changes Made
Fig. 1e	Fig. 1e	Updated data
Fig. 1f	Fig. 1f	Updated data
Fig. 3e	Fig. 3e	Updated data
-	Fig. 3h	New Data
Fig. 4c	Fig. S4d	Moved
Fig. 4d	Fig. 4c	Moved
Fig. 4e	Fig. 4d	Moved
-	Fig. 4f	New data
Fig. 4f	Fig. 4e	Moved + Updated data
-	Fig. 6g	New data
-	Fig. 7d	New analysis
Fig. 7d,e	Fig. 7e,f	Moved
Fig. 7f	Fig. 7g	Updated data

Fig. 7g	Fig. S7b	Moved + updated data
Fig. 8a	Fig. S8	Moved + updated data
Fig. 8b-d	Fig. 8a-c	Moved
Fig. 9b	Fig. S9b	Moved
Fig. 9c,d	Fig. 9b,e	Moved
-	Fig. 9c	New data
	Fig. 9d	New data
Fig. 9e	Fig. 9f	Moved
Fig. 9f	Fig. 9g	Moved + updated data
Fig. 9g	Fig. 9h	Moved
Table S1	Table S1	Updated data
Fig. S3	Fig. S3a	Moved
-	Fig. S3b	New data
-	Fig. S3c,d	New data + analysis
-	Fig. S4e	New data
Fig. S5b	Fig. S5b	Updated data
-	Fig. S6e	New data
Fig. S6e	Fig. S6f	Moved + updated data
Fig. S7a	Fig. S7a	Updated data
-	Fig. S7c,d	New data + analysis
Fig. S8a-g	Fig. S9b-h	Moved

REVIEWER COMMENTS

Reviewer #1

The manuscript by Munson and co-authors represents an effort to describe the role of cyclin G-associated kinase GAK and protein kinase C delta PRKCD in PRKN-independent mitophagy. The authors showed the importance of the above mentioned kinases both in vitro and in vivo, in their manuscript. The manuscript shows interesting results, however my comments will be related to my domain of expertise.

We thank the reviewer for their interest in our data and manuscript.

Question: Mass Spectrometry

It is unclear to me how much total proteins were used in the mass spectrometry experiment and how was the experiment designed.

We apologise for lack of clarity surrounding the mass spectrometry experiments design and execution. Cells were treated with indicated treatments before being moved on to ice and gently scraped in PBS to collect protein pellets (centrifugation 500xg / 5mins / 4°C). Cell pellets were lysed in RIPA buffer containing Halt™ protease and phosphatase inhibitor cocktail (ThermoFisher Scientific, #78446) and further homogenised with a sonicator (30 sec x 3 times with 30 sec interval), insoluble material was removed by centrifugation. Protein concentrations were determined by BCA assay (Pierce). For each replicate, 600 µg or 30 µg of protein was used for phosphoproteomic analysis or proteomic analysis respectively. Samples were reduced, alkylated and further digested with trypsin by FASP (Filter aided sample preparation) method. We have now included this information in the appropriate methods section (Line #803-808).

Authors say here: 'For each replicate (#?, if this is 2 like the Figure 8 suggests, this is not enough, at least 3-4 biological replicates are needed, in order to talk about the abundance changes, and p-values).

We thank the reviewer for their suggestion to improve the reliability of the data presented here. As a result, we have now carried out an additional 2x biological replicates (each carried out in technical duplicates), resulting in the final dataset now coming from 4x biological replicates, with each replicate comprised of technical duplicates. We have updated Fig. 1e-f, 4f, 8a (now Fig. S8) and supplementary table 1 to represent the updated dataset values, whilst numerical values have changed, these figures retain their original conclusions (Data are available via ProteomeXchange with identifier PXD022773).

600 µg of protein samples for phosphoproteomics and 30 µg for whole cell lysate proteomics were reduced..' Authors should modify this part, and better explain their idea, as there is no such thing as 'whole cell lysate proteomics'. Also, when authors say 'Cells were dissolved' this is incorrect, as cell cannot be dissolved in RIPA buffer, but the proteins can be lysed. So the authors could say: After cell lysis with RIPA buffer, proteins were measured with BCA kit. For the proteomic analysis of the whole cell lysate, this much proteins were used. For the proteomic analysis of the phosphoproteins from the cell lysate, this much proteins were used. Otherwise, it sounds confusing.

We apologise for the lack of clarity and confusion in this section of the methods, we have therefore modified the section as below (Line #803-815) to improve the text and address all of the reviewers points:

"Cells were lysed in RIPA buffer containing Halt™ protease and phosphatase inhibitor cocktail (ThermoFisher Scientific, #78446) and further homogenised with a sonicator (30 sec x 3 times with 30 sec interval), insoluble material was removed by centrifugation. Protein concentrations were determined by BCA assay (Pierce). For each replicate, 600 µg or 30 µg of protein was used for phosphoproteomic analysis or proteomic analysis respectively. Samples were reduced, alkylated and further digested with trypsin by FASP (Filter aided sample preparation) method. Dried peptide samples (from 600 µg or 30 µg of starting material for phospho-peptide or proteome respectively) were dissolved in 10 µl 0.1 % formic buffer and 30% of the final yield was loaded for each MS analysis."

Another important thing is whether authors used any kind of protease and phosphatase inhibitors in their RIPA buffer.

We apologise for this omission, Halt™ Protease and Phosphatase Inhibitor cocktail was included with RIPA buffer during cell lysis. We have now included this information in the appropriate methods section (Line #803-805):

"Cells were lysed in RIPA buffer containing Halt™ protease and phosphatase inhibitor cocktail and further homogenised with a sonicator (30 sec x 3 times with 30 sec interval), insoluble material was removed by centrifugation"

Also, authors should say how much peptides they loaded for LC-MS/MS, rather than saying 3 μ l.

We did not quantify the number of peptides after phospho-peptide enrichment to retain as much sample as possible for analysis. We utilised 600 μ g of protein as a starting point for phospho-peptide enrichment or 30 μ g for proteomic analysis. In both cases, 30% of the final yield of digested peptides were analysed per MS sample run. We have now clarified this in the methods (Line #813-815) as follows:

"Dried peptide samples (from 600 μ g or 30 μ g of starting material for phospho-peptide or proteome respectively) were dissolved in 10 μ l 0.1 % formic buffer and 30% of the final yield was loaded for each MS analysis"

Authors should provide details about the database used: how many entries, and what was the date the database was downloaded from Uniprot.

The database utilised was September_2018, we have now included this in the methods section (Line# 837-839):

"Minimal unique peptides were set to one, and FDR allowed was 0.01 (1 %) for peptide and protein identification. The Uniprot human database (September_2018) was used."

Reviewer #2

There is a major gap in our knowledge of the factors which drive basal mitophagy and hypoxia induced mitophagy. In this study, the authors use an image-based RNAi screen to identify lipid binding proteins which play a role in mitophagy, leading to the identification of the lipid binding kinases GAK and PRKCD. The authors show that GAK and PRKCD are specific for hypoxia induced mitophagy but not Parkin mediated mitophagy nor starvation induced autophagy. This was an interesting distinction which highlights the specificity of these kinases for a specific mitophagy pathway. Using animal models, the authors show that GAK and PRKCD play a role in basal mitophagy in vivo. In addition, while a partial mechanistic insight for a role for PRKCD in mitophagy was shown, the role played by GAK remains largely unclear. I don't think this is a major weakness for the study, since detailed mechanistic insights will likely be the focus of future studies, I do think it is important to provide a bit more of an understanding of how Gak and PRKCD drive mitophagy. For example, given the focus on lipid binding proteins in the screen, it would have been beneficial to determine whether lipid binding is necessary for the mitophagic roles of GAK and PRKCD. Additional suggestions to help partially address mechanism and strengthen the author's conclusions are provided in the comments below. Overall, this is an excellent study which makes a substantial and important advance to the fields of mitophagy and mitochondrial biology. The data are predominantly clear and convincingly support the author's conclusions. It is also clear that the discoveries made by Munson et al will be of interest to a broad readership and therefore highly suitable to the journal.

We thank the reviewer for the encouraging and helpful comments.

Specific comments:

1. Figure 3E: Can GAK colocalization with mitochondria during mitophagy be addressed via ectopic expression of the protein? Although the western blotting does not show evidence of mitochondrial localisation, it would be more convincing to address it via imaging since not all proteins which translocate to mitochondria are stably associated following isolation.

We have now ectopically expressed EGFP-GAK in U2OS cells and then treated cells (or not) with DFP, followed by co-staining with Mitotracker Red and live cell imaging to determine whether GAK might localize to mitochondria (New Fig. S3b). Transfected cells displayed a perinuclear accumulation of EGFP-GAK that possibly reflects earlier reports of Golgi localisation¹. We did not see any significant changes in GAK localisation following DFP treatment or any obvious co-localisation with mitochondria, however we noted the movement of EGFP-GAK positive structures on some occasions in close vicinity of mitochondrial networks (see still images from videos in New Fig. S3b).

2. Figure 4E: The western blot data are not entirely convincing given that even in controls there is little mitochondrial protein degradation. Can the authors quantify the results or alternatively try a longer DFP treatment (up to 48h)? There are typically much higher levels of mitophagy at 48h than at 24h DFP treatment and therefore the longer treatment will help to provide clearer results. Alternatively, the quantitative MS data in Figure 4F were very convincing, can this analysis also be conducted for PRKCD inhibition? This is particularly important for PRKCD since the western blot data in Figure 4E are not clear and show evidence of mitophagy occurring (MTCO2 degradation) even at high concentrations of sotrastaurin.

We thank the reviewer for this suggestion. In order to address this and a similar concern noted by reviewer #3 (point 7), we have now repeated western blots to quantify the effect of PKCi on mitochondrial protein loss induced by DFP after 24 and 48 hrs. Despite strong inhibition of Sotrastaurin on formation of red-only structures in the IMLS fluorescence analysis (Fig. 4a,b) and on citrate synthase activity (Fig. 4c) after 24 hrs of DFP treatment, we observe more variable loss of specific inner mitochondrial proteins (TIM23, MTCO2, COXIV) by western blot after 24 hrs DFP (old Fig. 4e, now Fig. 4d). We have now quantified the relative levels of TIM23, MTCO2, COXIV (relative to α -Tubulin loading control) after 24 or 48 hrs treatment with DFP in the presence of DMSO or Sotrastaurin (2-10 μ M). These data are now included as new Fig. 4f (48 hrs DFP) and new Fig. S4e (24 hrs DFP). Indeed, treatment with PKCi (sotrastaurin) and DFP led to a more reliable and significant inhibition of DFP-induced TIM23, MTCO2, COXIV degradation at 48 hrs than 24 hrs.

3. Figure 6: It is notable that LC3 is still lipidated (Figure 4E), especially with high concentrations of sotrastaurin treatment. Also, Figure 5G and 5E shows increased LC3 foci, indicating that sotrastaurin is affecting autophagy. To eliminate any off-target effects of the drug, and also to strengthen the authors conclusions, it would be beneficial to analyse Atg13 recruitment in GAK and PRKCD knockdown cells followed by rescue with an RNAi resistant construct.

We agree with the reviewer that sotrastaurin, especially at high concentrations, seems to induce LC3 lipidation and LC3 spot formation when treated cells were starved for 2 hrs in EBSS (Fig 5e-g). This increased effect of sotrastaurin on starvation-induced autophagy is however opposite of the inhibitory effect of sotrastaurin on DFP-induced mitophagy (Fig 4a-e).

We have attempted to rescue the mitophagy phenotype with siRNA resistant constructs of GAK and PRKCD, which unfortunately has turned out to be very difficult. We initially attempted to generate U2OS cells stably expressing siRNA resistant GAK and PRKCD constructs by Lentiviral transfection. We have generated siRNA resistant constructs of GAK and PRKCD that either have a V5 tag (for detection of GAK as there are no reliable antibodies) or untagged (to avoid potential unspecific localisation/function caused by tag, see details below). We have tried several rounds of a stable lentiviral-based integration approach (used routinely by the lab). Despite repeated transduction attempts, we found that transduced cells stopped proliferating making it extremely difficult to accumulate enough cells for experimental investigation (see also response to point 4). Thus, overexpression of both GAK and PRKCD appear to be problematic for cell viability, despite the lentiviral vectors having a relatively lower expressing (PGK) promoter. Impaired cell cycle regulation and cellular growth has been noted previously in several different cell models in response to PRKCD overexpression and may help explain our difficulties in generating these cell lines²⁻⁴.

We also tried to generate GAK and PRKCD KO cells using CRISPR/Cas9 and although this is another routinely used method in the lab, we were not able to select any viable clones functionally lacking these kinases (in the case of GAK this is perhaps to be expected given the earlier reported embryonic lethality in multiple model organisms⁵⁻⁷). Thus, we are unfortunately not able to provide any rescue experiments for GAK or PRKCD. We do however demonstrate an effect of both kinases on mitophagy using at least two different siRNA oligoes and multiple kinase inhibitors across several different mitophagy assays. Additionally, we have demonstrated

specificity with a zebrafish genetic *prkcd_ab* KO model and *gakh-1 C.elegans* model, both of which add significant weight to the importance of these kinases in mitophagy regulation.

Despite the problems with generation of rescue cell lines, we attempted to use what limited cells we could obtain for some experimental investigation. We were able to carry out a western blot to demonstrate overexpression of siRNA resistant PRKCD WT (New Fig. S3e):

We generated cells stably expressing both PRKCD WT and siRNA resistant PRKCD, in case the overexpression itself changed the response and to act as a more natural control. However, due to the high stable expression level of PRKCD, the siRNA treatment was no longer sufficient to reduce overall PRKCD levels to below that of endogenous PRKCD and no mitophagy impairment was observed. We believe this is due to the incomplete depletion of the protein allowing mitophagy to occur normally.

We subsequently analysed the effects of overexpression of PRKCD and carried out microscopy analysis of IMLS red dot formation +/- DFP in the overexpressed PRKCD cell line. We could still observe mitochondrial localisation (New Fig. S3c) and saw enhanced mitophagic flux (New Fig. S3d) and ULK1 Puncta formation (New Fig S6e), in line with expectations of a positive regulator of mitophagy.

We then attempted to examine the effect of siPRKCD on mitophagy and ATG13/ULK1 puncta formation using two different oligos in WT IMLS. PRKCD transcript levels were reduced by 60-70% as verified by qPCR (data included for reviewer only):

This was sufficient to reduce IMLS red dot formation by ~30-60% in line with earlier screening data. However, we did not observe a corresponding reduction in ULK1 puncta formation in the presence of DFP (data included here for reviewer).

Thus, the knockdown efficiency of PRKCD is either not sufficient to block assembly of this early autophagy complex or it could indicate that other PKC isozymes may be involved in mitochondrial recruitment of the ULK1 complex, as this is inhibited by the use of PKCi.

In addition, given that LC3B staining was shown in Figure 6B, can this analysis be extended to GAK and PRKCD inhibition? This analysis would help to address the idea put forward by the authors that GAK inhibition may reduce mitochondrial loading into autophagosomes (and help to justify the correlative mitochondrial morphology data). If this is not the case, then it would point toward an autophagosome-lysosome defect (which may correlate with the altered lysosome morphology data).

We have now examined the localisation of LC3 puncta following treatment with GAKi/PKCi and combined with DFP treatment. As expected, there is an increased co-localisation of LC3 with mitochondrial structures during DFP (New Fig. 6g). We see notable decreases in LC3-Mito overlap during DFP when co-treated with GAKi (Fig. 6g), suggesting that GAK activity is important for mitochondrial loading into autophagosomes. This effect size is perhaps not entirely sufficient to explain the full reduction in mitophagy that we observe using GAKi, but as we also see mitochondrial network abnormalities (Fig. 7a-c) and lysosomal abnormalities in GAKi treatment (Fig. 7e-f), these also likely contribute to reduced mitophagy efficacy. Moreover, the fact that Parkin-dependent mitophagy and starvation-induced autophagy proceeds normally in GAKi treated cells, argue against a generic autophagosome-lysosome defect.

It also might be worth the authors considering whether TFEB is correctly induced following GAK inhibition.

We thank the reviewer for the suggestion of examining TFEB localisation. We monitored the localisation of TFEB-mCherry following treatment +/- DFP in combination with DMSO/GAKi/GAKc or a Torin1 control. DFP treatment alone induced a mild, but not significant increase in nuclear TFEB localization (New Fig. S7c,d), while as expected, Torin1, caused a strong

nuclear TFEB accumulation. Interestingly, GAKi treatment gave a significant increase in the amount of nuclear TFEB (Fig. S7c,d), which likely explains the increased lysosomal abundance noted by LAMP1 staining in Fig. 7e-f. Furthermore, we observe notable inhibition of mTOR activity following GAKi and DFP combined treatment (Fig. S6e) that also likely contributes to TFEB localisation and LAMP1 expression differences. We have now included these points in the results section (Line #328-333).

4. Given the major focus on screening for lipid binding proteins, it would be important to address whether lipid binding by GAK and PRKCD is required for mitophagy. This could be achieved by expressing lipid binding mutants (or lipid domain deletion) in RNAi cells followed by mitophagy assessment.

We agree with the reviewer that it is important to address whether lipid binding by GAK and PRKCD is required for their role in regulation of mitophagy. Both kinases contain a C2 domain, that can interact with different lipids (and possibly also proteins) in a Ca^{2+} -independent manner while PRKCD additionally contains two C1 domains, predicted to interact and activate in response to diacylglycerol (DAG).

To address if and how protein lipid binding affects GAK/PRKCD function during mitophagy we took two approaches. First, as suggested by the reviewer, we attempted to generate stable cell lines expressing siRNA resistant WT or lipid domain deletion mutants of GAK and PRKCD. Second, we generated corresponding constructs for protein purification from HEK293 cells, followed by their lipid-binding assessment.

As described above, to generate stable cell lines expressing siRNA resistant versions of PRKCD/GAK WT or Δ lipid binding domain (Δ LBD) deletions, we cloned (by Gibson assembly) PRKCD/GAK WT or Δ LBD deletion mutants into pLenti-PGK-Puro-Dest vectors (sequence validated) and generated Lentiviral particles by co-transfecting with psPAX2 and VSVG expression vectors. Viral particles were harvested and concentrated with Lenti-X concentrator and used to transduce U2OS IMLS cells (MLS-mCherry-EGFP) for mitophagy analysis. Positively integrated cells were selected with puromycin and validated by death of non-transduced control cells. However, after successful selection, we encountered problems with cells entering senescent-like states that stopped actively proliferating. We have attempted to culture cells with conditioned media to boost growth rates and have repeated lentiviral transduction several times. However all rescued cell lines either died (all GAK transduced cells) or stopped dividing (in particular the PRKCD Δ LBD mutant), which severely limited follow up experiments. As noted earlier, cell cycle and growth disruption appears to be associated previously with PRKCD overexpression⁵⁻⁷.

We did however, after several months, manage to obtain some dividing cells of PRKCD WT and Δ LBD mutant that were used for experiments. Unfortunately, the PRKCD Δ LBD mutant cell line had lost most of the IMLS expression, which made it difficult to analyse effects on mitophagy (using the red-only IMLS assay). We did however manage to examine mitochondrial localisation of the PRKCD WT and Δ LBD mutant protein using mitochondrial fractionation, followed by western blotting with an anti-PRKCD antibody. To our surprise, we could clearly identify the PRKCD Δ LBD protein in the mitochondrial fraction (New Fig. 3h), suggesting that mitochondrial

localisation of PRKCD is independent of its C1 or C2 domain. Whilst lipids such as DAG may still play a role in regulating the activity of PRKCD, it may be a protein-protein interaction facilitated by another part of the PRKCD protein that is more critical for determining its mitochondrial localisation.

As lipid binding of PRKCD seems dispensable for its localisation to mitochondria, we did not follow up further our initial attempts to address the lipid binding specificity of the GAK and PRKCD C1/C2 domains. As shown below (data included for reviewer only), we generated FLAG-PRKCD WT and FLAG-PRKCD Δ C2- Δ C1 along with FLAG-GAK WT and FLAG-GAK Δ PD (Phosphatase domain) + Δ C2 constructs for expression in HEK293 cells, followed by analysis of their lipid binding preferences using lipid overlay strips (Echelon biosciences). Proteins were expressed in HEK293 cells and affinity purified by FLAG resin, as demonstrated by Coomassie blue staining.

Purified proteins were then incubated with lipid strips before subsequent immunodetection of bound proteins:

Surprisingly, only binding to phosphatidic acid (PA) and minor PI(3)P binding was observed across constructs, however no loss of binding was seen by deletion of the lipid binding domains. Expression of GST-Phosphatase Domain alone (from GAK) was previously shown to bind multiple phosphoinositide species including PI(3)P, PI(4)P and PI(5)P⁸. However, we are unable to recapitulate this phenotype when working with GAK protein, which may indicate differences in the phosphatase domain accessibility to lipids when present within the full protein.

PRKCD is a novel PKC isoform that is known to bind DAG in a calcium-independent manner via the C2 domain, however this interaction is complex with additional lipid interactions required to facilitate successful binding and thereby difficult to recapitulate via lipid overlay⁹. Additionally, proteins were isolated from unstimulated cells, potentially missing relevant post-translational modifications or interaction partners.

5. Sup Fig 7: The GAKi image has barely visible mitochondria as compared to the other samples. Can the authors clarify how mitochondrial network classification was conducted on those kinds of images?

We apologise for the original image with weakly intense mitochondrial staining for some of the treatments. We have now updated the figure panel with images that more accurately represent the mitochondrial morphologies from different cellular treatments (Supplementary fig. 7a).

To classify mitochondrial phenotypes, mitochondrial features were first identified in CellProfiler from EGFP images using the U2OS IMLS cell line. Subsequently, CellProfiler Analyst was used to generate two classification bins based upon siDRP1 or siOPA1 treated samples as training sets (to represent fragmented or hyperfused phenotypes, respectively). Key features were identified using a random forest model and led to a confusion matrix of >0.90 for each phenotype. These classification parameter were then applied to all samples and cells scored for hyperfused or

fragmented phenotype as demonstrated in Fig. 7b. We have updated the methods section to reflect this more detailed methodological description (Line #709-716):

"To classify mitochondrial phenotypes, mitochondrial features were first identified in CellProfiler (v2.8.0, The Broad Institute) from EGFP images using the U2OS IMLS cell line. Subsequently, CellProfiler Analyst (v2.2.1, The Broad Institute) was used to generate two classification bins based upon siDRP1 or siOPA1 treated samples as training sets (to represent fragmented or hyperfused phenotypes respectively). Key features were identified using a random forest model and led to a confusion matrix of >0.90 for each phenotype. These classification parameters were then applied to all samples and cells scored for hyperfused or fragmented phenotype and plotted as a percentage of the overall population."

Minor comments:

1. Line 47: The idea that Parkin mitophagy is not relevant *in vivo* due to it not being involved in basal mitophagy is not entirely accurate since it is largely a stress response pathway. Given this, it is unclear why there would be an expectation that Parkin would drive mitophagy under basal, non-stressed conditions. There are numerous studies showing activation of Parkin mitophagy during stress in animal models e.g. alcohol induced liver disease (PMID: 26159696), renal ischemia-reperfusion injury (PMID: 29172924), exercise (PMID: 29812989), myocardial infarction (PMID: 23152496). It would be worthwhile clarifying this statement.

Upon reading line 47 again we agree with the reviewer that our statement inadvertently downplayed the existing *in vivo* linkage of Parkin to mitophagy by focusing solely upon basal mitophagy where it is not thought to play a role. We have therefore revised the text (now line# 52-55) to read.

"Whilst PRKN function has been linked to regulation of stress-induced mitophagy *in vivo* (e.g. exhaustive exercise⁴, alcohol-induced liver disease⁵, myocardial infarction⁶), basal mitophagy *in vivo* appears to occur largely independent of PINK1/PRKN. This has been demonstrated across mice, fly, and zebrafish models⁷⁻⁹"

2. Figure 1: The authors may wish to consider moving the data in Figure 1 to the supplement since it doesn't add to the study's main conclusions and mainly confirms the well-established paradigm of DFP induced mitophagy and the tandem fluorescent reporter as a measure of mitophagy. In addition, the data in Figure 8A, while certainly useful as a resource, were not used directly to follow up on and address the GAK inhibition mitophagy defect and would therefore be best placed in the supplement.

Whilst we appreciate that Fig. 1 introduces the previously established concept of DFP induction of mitophagy, we believe it adds tremendous value to the reader as an immediate overview of the potential methods to monitor this form of mitophagy that are utilised throughout the manuscript. In addition, CLEM data and Mass spec analysis with subsequent GO annotation offer orthogonal readouts that have not been applied in this manner before to our knowledge for DFP-induced mitophagy and adds significant validation of our system for measuring mitophagy using our slightly modified matrix-localised mitochondrial reporter (compared to the mito-QC outer membrane approach). We have therefore chosen to retain Figure 1 in the original

position, however, we have condensed some text sections since aspects are well established in the field.

We agree with the reviewer that the GAK proteomics data were not followed up and for that reason have now moved it to Fig S8.

3. The authors may wish to avoid repetitively using descriptions such as ‘interestingly’ and ‘interesting’ and ‘disappointing’, sometimes it is best to leave it to the readers to decide.

We agree with the reviewer and have now modified the text accordingly.

Reviewer #3 (Remarks to the Author):

The manuscript “GAK and PRKCD are positive regulators of PRKN-independent mitophagy” by Munson et al. describes the results from a siRNA screen targeting lipid domain containing proteins with regards to a possible function in mitophagy. The paper describes the initial results from the screen, validation of hits – GAK and PRKCD – by small molecule inhibitors and an assessment of their potential function in mitophagy in vivo, using *C. elegans* and zebrafish as model systems. Overall, this is a very interesting paper that describes novel insights into parkin-independent mitophagy and the function of GAK and PRKCD at the molecular level. The paper is very well written and the experimental data is of high standard.

We thank the reviewer for the positive and helpful comments.

I have the following concerns that should be addressed.

1. I wonder if the screen of 197 genes was carried out in the presence and also in the absence of DFP. The comment “data not shown” in line 116 seems to suggest so. It would be important to also show the screen results without DFP treatment as they might hint at basal mitophagy regulators.

The basal mitophagy level without any stimulus in the U2OS IMLS cell line (matrix localized MLS-mCherry-EGFP) is extremely low, as analysed by quantification of red-only puncta (e.g Fig 1b), and differs significantly from the *in vivo* situation. Whilst we agree it would be nice to identify basal mitophagy regulators in this way, practically, the background mitophagy is too low to observe down regulation after siRNA treatment even with positive controls such as siULK1. This can be seen in the graphs below (included for reviewer only) showing the raw number of red only structures present per cell +/- DFP:

We carried out initial screening +/- DFP treatment to potentially capture spontaneous inducers of mitophagy, however none were initially observed and therefore we did not follow this up further in a robust, presentable, manner. For that reason we have excluded this data from the manuscript and we have removed the text comment of "data not shown".

2. In the analysis of mitophagy or red only puncta, is there any consideration of how cell numbers might affect the readout?

For experiments involving imaging and red dot readouts, the mitophagy values have been normalised to the non-targeting siRNA (siNT) to reflect an average on a per cell basis. This allows the measure to be independent of cell number. We apologise that this had not been made clear and have updated the graphs y-axis in Fig. 2 to read "Mitophagy per cell (normalised to siNT control)".

3. In supplementary figure 1, there is a high variability between the secondary and tertiary screen. While I appreciate that this can be the case for siRNA, it is still surprising that as little as twice the siRNA concentration can result in such great changes in variability. Is there any possible explanation for this? Does this also mean that potentially many hits were missed from the primary screen?

We agree that unfortunately there is a certain degree of variability obtained when carrying out siRNA screens, further highlighting how important it is to robustly evaluate candidates through a variety of follow up approaches (as done in this manuscript). It is worth noting that several potentially significant experimental changes were introduced between the second and tertiary screen beyond transfected oligonucleotide concentration. These include a different geographical location and author running the protocol that resulted in modifications to the microscope utilised, imaging plate type, cell number per transfection, FBS batch, treatment time post-

transfection and differences in transfection media. In light of all these changes, we believe a certain variability is unfortunately to be expected. In any case, we have plotted results from targets common between the two screens (included here for reviewers only). This demonstrates a moderate linear correlation between the screens with an R^2 value of 0.623 (shown below). We then only chose significant candidates that displayed responses from 2x oligos as a shortlist presented in Fig. 3a.

4. In Figure 3a: are significances missing for siULK1? It seems highly significant, but there is no statistical annotation.

We thank the reviewer for spotting this omission. We have now correctly updated the graph with significance stars for siULK1 in Fig. 3a.

5. What is the dotted line in Figure 3b? Not explained in the legend.

We apologise for the lack of explanation. The red and orange dashed lines are intended to highlight the average % of interacting proteins based upon the "All Screen" values. Bars above this value indicate higher than average for equivalent proteins in the screen. We have now updated the legend for Fig. 3b as follows:

"b Protein-protein interaction networks for candidate proteins (see methods) were plotted by % of interacting proteins belonging to each highlighted compartment, value in brackets represents total number of interacting proteins. Dashed lines indicate average values from all screened proteins for mitochondria (red) or autophagosome (orange)."

6. The K_d values in supplementary figure 4a: were these measured in this study or is this referring to another publication?

We apologise for any confusion surrounding this point. The Kd values are reported from previous studies^{10,11}. We have now clarified the legend for Fig. S4a with the following:

"a Structural comparison of different GAK inhibitors utilised in this study and their Kd/Ki values reported in initial publications [29,30]"

7. It would be useful to see a quantification of the blot in Figure 4e, especially for the three genes that changed: MTCO2, COXIV, TIM23

As suggested by the reviewer (and a similar request by reviewer #2, point 2), we have now repeated the western blots to quantify the effect of PKCi on mitochondrial protein (TIM23, MTCO2, COXIV) loss induced by DFP. Despite strong inhibition of red-structures in the IMLS analysis (Fig. 4a-b) or citrate synthase activity (now Fig. 4c), we saw less pronounced loss of specific mitochondrial proteins (including TIM23, MTCO2, COXIV) by western blot at 24 hrs (New Fig. S4e). It is unclear why we observe such differences between markers.

Following a recommendation from Reviewer #2, we also repeated experiments with 48 hrs of DFP treatment in the absence or presence of PKCi (sotrastaurin), followed by immunoblotting for TIM23, MTCO2 and COXIV and quantification of their protein levels (New Fig. 4f). The DFP-induced loss of mitochondrial proteins was greater at 48 hrs than at 24 hrs and co-treatment with PKCi (sotrastaurin) led to a more consistent and significant inhibition of TIM23, MTCO2, COXIV protein degradation compared to 24 hrs.

8. Supplementary figure 5b is missing a control that shows a shift by Phos-tag.

We have now repeated the experiment and updated Fig. S5b with calf intestinal alkaline phosphatase (CIP) treatment as a control to remove phosphorylated residues. With this, we can verify that the uppermost bands are sensitive to CIP treatment and thereby represent phosphorylated BNIP3 (P-BNIP3) and BNIP3L (P-BNIP3L).

9. Supplementary figure 6e is missing a blot for total ULK1; and is missing a control that changes the phosphorylation of ULK1.

We have now repeated the experiment with total ULK1 control and include the mTOR inhibitor Torin1 as a positive control along with p70 S6K as an additional mTORC1 substrate readout. This data (new Fig. S6e) demonstrates significant inhibition of mTOR following combination treatment with GAKi and DFP.

10. There is quite a bit of follow-up work with a rather non-selective PKC inhibitor. Some of this could be further supported by siRNA knockdown of PRKCD. Otherwise, one might want to tone down any conclusions about specific effects by PRKCD. This is well done in the discussion section, but could be clearer in the results.

We agree that unfortunately there are limited tools for dissecting PKC family member function apart from one another, along with potential compensatory redundancy of isoforms if carrying out stable modifications of cell lines. siRNA knockdown of PRKCD was significant for reducing

mitophagy by IMLS fluorescence and citrate synthase activity, however we also saw reduced mitophagy when targeting other novel PKC isoforms by siRNA (Supplementary Fig. 4d). The most potent inhibition of mitophagy was observed using chemical inhibitors, like sotrastaurin, that inhibit several PKC isoforms, potentially indicating an additive effect of inhibiting multiple PKC kinases with partially redundant roles. Due to problems with generation of PRKCD specific rescue cells (see our response to comments 3 and 4 from reviewer #2), we have toned down the conclusions in general throughout the results section and have attempted to highlight further that there could be a broader effect of novel PKC isozymes.

Nevertheless, we believe our data strongly support a role for PRKCD in mitophagy. We have robustly shown that i) PRKCD localises to mitochondria (by immunofluorescence and mitochondrial fractionation), ii) mitophagy is inhibited using several different oligoes against PRKCD (in addition to two PKCi) across different mitophagy assays, iii) mitophagy is inhibited under both basal and DMOG-induced conditions using a zebrafish genetic *prkcd_ab* KO model.

11. Quantification of Figure 7c is needed to interpret the data.

We agree with the reviewer and have now included quantification for this figure, the data is now shown in Fig. 7d. These experiments were carried out in PRKN negative cells, CCCP therefore leads to visual fragmentation of mitochondrial networks without induction of mitophagy as this requires PRKN expression. The addition of CCCP to force mitochondrial fragmentation was however not sufficient to rescue mitophagy (IMLS red dot formation) during DFP + GAKi co-treatment.

12. It seems to me, the quantification of 7e in 7f is not matching. There is an increase in control-GAKi compared to DMSO, which is not shown in the quantification.

We thank the reviewer for their keen observations – we have spent time closely examining all replicates and troubleshooting the analysis pipeline utilised for quantification of LAMP1 structures. We identified some issues with object separation in close proximity structures and filtering of rounded non-viable cells present in the GAKi treatment – we have now updated the graph (new Fig. 7g) with the updated analysis and additional experimental data that indicates increased numbers of LAMP1 structures in the GAKi treated samples both +/- DFP treatment. This is also in agreement with increased nuclear TFEB translocation in GAKi treated cells compared to DMSO treatment (new Fig. S7c-d).

13. Not sure what Figure 7g shows – it is missing an untreated control and quantification. Might be possible to delete?

We apologise for the lack of clarity surrounding this figure. Figure 7g is intended to demonstrate that during GAKi treatment lysosomes are still able to accumulate lysotracker stain, thereby an indication that lysosomal acidification is still occurring following GAKi treatment. We have now updated the image panel to include a DMSO control image and these data have now been moved to Fig. S7b.

14. It is not clear how Figure 8c was calculated. Does n=25 means mitochondria or cells? There are some very large blobs and some small puncta – was any threshold applied? Which ones were counted – large blobs and also puncta or just puncta?

For Fig. 8c, n= 25 represents the total number of worms analysed per group (control or gakh-1 siRNA). For analysis, thresholds were set to detect mitochondrial structures without size restrictions, therefore both small and larger structures were part of the analysis. We have updated the figure legend with the following "n= 25 worms per group" and the methods section (Line #738-741) with:

"Intensity thresholds were set to detect mitochondrial structures without size restrictions. Average pixel intensity values and number of GFP/DsRed puncta were calculated by image analysis of different animals using FIJI."

References:

1. Kametaka, S. *et al.* Canonical interaction of cyclin G associated kinase with adaptor protein 1 regulates lysosomal enzyme sorting. *Mol. Biol. Cell* **18**, 2991–3001 (2007).
2. Mischak, H. *et al.* Overexpression of protein kinase C- δ and - ϵ in NIH 3T3 cells induces opposite effects on growth, morphology, anchorage dependence, and tumorigenicity. *J. Biol. Chem.* **268**, 6090–6096 (1993).
3. Fukumoto, S. *et al.* Protein kinase C δ inhibits the proliferation of vascular smooth muscle cells by suppressing G1 cyclin expression. *J. Biol. Chem.* **272**, 13816–13822 (1997).
4. Watanabe, T. *et al.* Cell division arrest induced by phorbol ester in CHO cells overexpressing protein kinase C- δ subspecies. *Proc. Natl. Acad. Sci. U. S. A.* **89**, 10159–10163 (1992).
5. Lee, D.-W., Zhao, X., Yim, Y.-I., Eisenberg, E. & Greene, L. E. Essential role of cyclin-G-associated kinase (Auxilin-2) in developing and mature mice. *Mol. Biol. Cell* **19**, 2766–76 (2008).
6. Kandachar, V., Bai, T. & Chang, H. C. The clathrin-binding motif and the J-domain of Drosophila Auxilin are essential for facilitating Notch ligand endocytosis. *BMC Dev. Biol.* **8**, 1–15 (2008).
7. Greener, T. *et al.* Caenorhabditis elegans auxilin: A J-domain protein essential for clathrin-mediated endocytosis in vivo. *Nat. Cell Biol.* **3**, 215–219 (2001).
8. Lee, D., Wu, X., Eisenberg, E. & Greene, L. E. Recruitment dynamics of GAK and auxilin to clathrin-coated pits during endocytosis. *J. Cell Sci.* **119**, 3502–12 (2006).
9. Stahelin, R. V *et al.* Diacylglycerol-induced membrane targeting and activation of protein kinase Cepsilon: mechanistic differences between protein kinases Cdelta and Cepsilon. *J. Biol. Chem.* **280**, 19784–93 (2005).
10. Pu, S. Y. *et al.* Optimization of Isothiazolo[4,3- b]pyridine-Based Inhibitors of Cyclin G Associated Kinase (GAK) with Broad-Spectrum Antiviral Activity. *J. Med. Chem.* **61**, 6178–6192 (2018).
11. Asquith, C. R. M. *et al.* SGC-GAK-1: A Chemical Probe for Cyclin G Associated Kinase (GAK). *J. Med. Chem.* **62**, 2830–2836 (2019).

REVIEWERS' COMMENTS

Reviewer #1 (Remarks to the Author):

I appreciate the hard work done to address all of my concerns, I believe the manuscript has gone through a significant improvement.

Reviewer #2 (Remarks to the Author):

The authors have clearly gone to extensive effort to address the reviewer comments. Some points could not be completely addressed due to experimental challenges, but despite this, the manuscript is greatly improved with strengthened conclusions and also has the addition of some interesting new results. I have no further comments to add, congratulations on a very nice study.

Reviewer #3 (Remarks to the Author):

Thank you for addressing my comments! I appreciate that rescue experiments for siRNA knockdown can often be difficult, for various reasons. The authors have done a thorough job in taking the comments into consideration and I am happy with their response. I recommend publication.

REVIEWERS' COMMENTS

We would like to thank all reviewers for their time and critical input to the manuscript revision, we believe this process has strengthened the manuscript greatly.

Reviewer #1 (Remarks to the Author):

I appreciate the hard work done to address all of my concerns, I believe the manuscript has gone through a significant improvement.

We thank the reviewer for their positive comments.

Reviewer #2 (Remarks to the Author):

The authors have clearly gone to extensive effort to address the reviewer comments. Some points could not be completely addressed due to experimental challenges, but despite this, the manuscript is greatly improved with strengthened conclusions and also has the addition of some interesting new results. I have no further comments to add, congratulations on a very nice study.

We appreciate the reviewers understanding of the experimental challenges faced and thank the reviewer for their positive comments on the study.

Reviewer #3 (Remarks to the Author):

Thank you for addressing my comments! I appreciate that rescue experiments for siRNA knockdown can often be difficult, for various reasons. The authors have done a thorough job in taking the comments into consideration and I am happy with their response. I recommend publication.

We appreciate the understanding of the reviewer regarding the technical challenges of siRNA rescue experiments and thank the reviewer for their positive opinion of the manuscript.